# Matching the Optimal Denoiser in Point Cloud Diffusion with (Improved) Rotational Alignment

## Abstract

Diffusion models are a popular class of generative models trained to reverse a noising process starting from a target data distribution. Training a diffusion model consists of learning how to denoise noisy samples at different noise levels. When training diffusion models for point clouds such as molecules and proteins, there is often no canonical orientation that can be assigned. To capture this symmetry, the true data samples are often augmented by transforming them with random rotations sampled uniformly over $SO(3)$. Then, the denoised predictions are often rotationally aligned via the Kabsch-Umeyama algorithm to the ground truth samples before computing the loss. However, the effect of this alignment step has not been well studied. Here, we show that the optimal denoiser can be expressed in terms of a matrix Fisher distribution over $SO(3)$. Alignment corresponds to sampling the mode of this distribution, and turns out to be the zeroth order approximation for small noise levels, explaining its effectiveness. We build on this perspective to derive better approximators to the optimal denoiser in the limit of small noise. Our experiments highlight that alignment is often a 'good enough' approximation for the noise levels that matter most for training diffusion models.

## 1 Introduction

Diffusion-based generative models have emerged as a powerful class of generative models for complex distributions in high-dimensional spaces, such as natural images and videos.

Diffusion models operate on the principle of reversing a noising process by *learning how to denoise*. Let $p_x$ be our data distribution defined over $\mathbb{R}^d$. Let $p_y(\cdot; \sigma)$ be the distribution of $y = x + \sigma\eta$ where $x \sim p_x, \eta \sim \mathcal{N}(0, \mathbb{I}_d)$. $y$ represents a noisy sample at a particular noise level $\sigma$. When the noise level is zero, we have that $p_y(\cdot; \sigma = 0) = p_x$, as expected. On the other end, as $\sigma \to \infty$, the data distribution $p_x$ is effectively wiped out by the noise and $p_y(\cdot; \sigma) \to \mathcal{N}(0, \sigma^2 \mathbb{I}_d)$. As explained by Karras et al. (2022), the idea of diffusion models is to randomly sample an initial noisy sample $y_M \sim \mathcal{N}(0, \sigma_M^2 \mathbb{I}_d)$, where $\sigma_M$ is some large enough noise level, and sequentially denoise it into samples $y_i$ with noise levels $\sigma_M > \sigma_{M-1} > \cdots > \sigma_0 = 0$ so that at each noise level $y_i \sim p(y; \sigma_i)$. Assuming the denoising process has no error, the endpoint $y_0$ of this process will be distributed according to $p_x$.

Many schemes (Song et al., 2022; Yang et al., 2024; Karras et al., 2022) have been built for sampling diffusion models; the details of which are not relevant here. The essential component across these is that one trains a denoiser model $D$ that minimizes the denoising loss $l_{\text{denoise}}$ across noisy samples at a range of noise levels:

$$\min_D l_{\text{denoise}}(D) = \min_D \mathbb{E}_{x \sim p_x} \mathbb{E}_{\eta \sim p_\eta} \mathbb{E}_{\sigma \sim p_\sigma} [\|D(x + \sigma\eta; \sigma) - x\|^2] \tag{1}$$

Once learned, the denoiser model $D$ is then used to iteratively sample the next $y_i$ from $y_{i+1}$ using a numerical integration scheme, such as the DDIM update rule (Song et al., 2022) (for example):

$$y_i = y_{i+1} + \left(1 - \frac{\sigma_i}{\sigma_{i+1}}\right) (D(y_{i+1}, \sigma_{i+1}) - y_{i+1}) \tag{2}$$

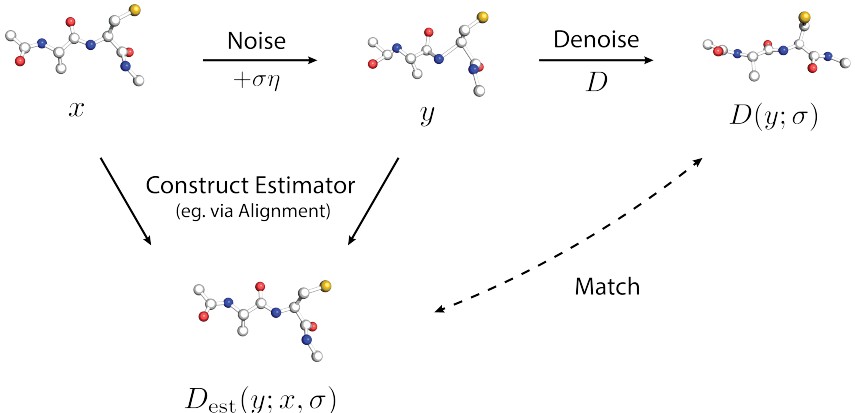

Figure 1: Overview of the training process of a denoising diffusion model, represented by $D$. A sample point cloud $x$ is first noised to give $y$. $D$ denoises $y$ to give a new point cloud $D(y; \sigma)$, which gets matched to an estimator $D_{\text{est}}(y; x, \sigma)$ of the optimal denoiser $D^*$. The usual estimator is $D_{\text{est}}(y; x, \sigma) = x$. Here, we show that rotational alignment gives rise to better estimators of $D^*$.

We are particularly interested in the setting where the data samples naturally live in three-dimensional space, such as molecular conformations, protein structures and point clouds. Often in these settings, there is no canonical 3D orientation that can be assigned to the data samples. Thus, we would like to sample all orientations of the data samples with equal probability. Formally, this means that our 'true' data distribution is $\text{Aug}[p_x]$, where each sample $x \sim p_x$ has been augmented with uniformly sampled rotations $\mathbf{R} \in SO(3)$.

There are two main approaches to obtain this goal of making the sampled distribution $SO(3)$-invariant (at least approximately): 1) *learning an $SO(3)$-equivariant denoiser $D$* or 2) *learning with rotational augmentation*.

Furthermore, to enforce this rotational symmetry, it is common (Xu et al., 2022; Abramson et al., 2024; Wohlwend et al., 2024; Daigavane et al., 2025; Klein et al., 2023b; Dunn & Koes, 2025) to perform an alignment step (either between $y$ and $x$, or between $D(y; \sigma)$ and $x$) with the Kabsch-Umeyama algorithms (Kabsch, 1976; Umeyama, 1991) before computing the loss of denoising. However, not much is known about the effect of this alignment step. In particular, does alignment introduce bias in the learning objective? If so, can we improve the alignment operation to reduce this bias? Our paper answers these key questions, and is organized as follows:

- In Section 3, we derive the optimal denoiser for point cloud diffusion. Importantly this denoiser separates into an expectation of the optimal augmented single sample denoiser.

- In Section 4, we show that training a diffusion model is equivalent to matching the single sample optimal denoiser, motivating the construction of better estimators of the optimal denoiser.

- In Section 6, we show the optimal single sample denoiser involves an expectation of a matrix Fisher distribution over $SO(3)$ and rotation alignment corresponds to a zeroth order approximation using the mode of this distribution.

- In Section 7, we build on this insight to obtain better estimators of the optimal denoiser by approximating the expectation via Laplace's method. These estimators enjoy reduced bias relative to the standard alignment based estimator, at *no additional computational cost*, with numerical evidence in Section 8.

- In Section 9, we experiment with these improved estimators to train better diffusion models. At lower noise levels, the effective improvement from using these higher-order correction terms is minimal, suggesting that alignment is already a good enough approximation.

## 2    PROBLEM SETUP

**Point Clouds**: Let $x \in \mathbb{R}^{N \times 3}$ represent a point cloud with $N$ points living in 3D space. The entries in $x$ denote the Cartesian coordinates of each point in the point cloud; each row $x_i^\top$ is the vector of 3D coordinates of the $i$th point. We define $\|x\|^2 \equiv \sum_{i=1}^{N} \|x_i\|^2$.

Often, the point clouds are associated with some $SO(3)$-invariant features (eg. atomic numbers or charges for atoms in a molecule). These features may be sampled a priori and provided to the denoiser, or undergo their own denoising process with a separate denoiser model (Hoogeboom et al., 2022; Yim et al., 2023; 2024; Campbell et al., 2024). Our analysis is not affected in either case, so we omit these features from further discussion.

**Rotations**: $SO(3)$ refers to the group of rotations in three dimensions. Since we are working with point clouds which are sets of vectors, it is natural to consider the representation of rotations as rotation matrices $\mathbf{R} \in \mathbb{R}^{3 \times 3}$. Thus, the action of a rotation $\mathbf{R}$ on the point cloud $x = [x_i^\top]_{i=1}^{N}$ is $\mathbf{R} \circ x$ obtained by rotating the coordinates of each point by $\mathbf{R}$ independently:

$$\mathbf{R} \circ x \equiv x\mathbf{R}^\top \equiv [x_i^\top \mathbf{R}^\top]_{i=1}^{N}. \tag{3}$$

Note that rotation matrices are orthonormal: $\mathbf{R}^\top \mathbf{R} = \mathbb{I}_3$. Further, the group action is associative:

$$\mathbf{R}_1 \mathbf{R}_2 \circ x = \mathbf{R}_1 \circ (\mathbf{R}_2 \circ x). \tag{4}$$

$SO(3)$-**Invariance**: Let $p$ be a distribution over $\mathbb{R}^{N \times 3}$. We say that $p$ is $SO(3)$-invariant if:

$$p(\mathbf{R} \circ x) = p(x) \quad \text{for all } \mathbf{R} \in SO(3), x \in \mathbb{R}^{N \times 3}. \tag{5}$$

Since there is no way to appropriately normalize a translation-invariant $p_x$, we center point clouds $x$ such that $\sum_{i=1}^{N} x_i = \vec{0}$, as is commonly done in the literature. This operation does not change our analysis.

**Rotational Augmentation**: The simple (yet approximate) approach to obtain an equivariant distribution is to augment the data distribution $p_x$ with randomly sampled rotations. In particular, we sample $x \sim p_x$ from our data distribution $p_x$, $\mathbf{R} \sim u_{\mathbf{R}}$ from the uniform distribution $u_{\mathbf{R}}$ over $SO(3)$ as defined by the Haar measure, and return $\mathbf{R} \circ x \sim \text{Aug}[p_x]$, which is defined as:

$$\text{Aug}[p_x](x') = \int_{SO(3)} p_x(\mathbf{R}^{-1} \circ x') u_{\mathbf{R}}(\mathbf{R}) d\mathbf{R}. \tag{6}$$

By construction, $\text{Aug}[p_x](\mathbf{R} \circ x) = p_x(x) u_{\mathbf{R}}(\mathbf{R})$ for any $\mathbf{R}$. A simple proof (Appendix A.2) shows that $\text{Aug}[p_x]$ is always $SO(3)$-invariant. When training with rotational augmentation,[1] the loss becomes:

$$\min_D l_{\text{aug}}(D) = \min_D \mathbb{E}_{\mathbf{R} \sim u_{\mathbf{R}}} \mathbb{E}_{x \sim p_x} \mathbb{E}_{\eta \sim p_\eta} \mathbb{E}_{\sigma \sim p_\sigma} [\|D(\mathbf{R} \circ (x + \sigma\eta); \sigma) - \mathbf{R} \circ x\|^2]. \tag{7}$$

$SO(3)$-**Equivariant Denoisers**: An $SO(3)$-equivariant denoiser $D$ commutes with all rotations $\mathbf{R}$. Formally:

$$D(\mathbf{R} \circ (x + \sigma\eta)) = \mathbf{R} \circ D(x + \sigma\eta) \tag{8}$$

for all point clouds $x$, noise $\eta$, noise levels $\sigma$ and rotations $\mathbf{R}$.

In Appendix A.1, we show that, given a $SO(3)$-invariant initial distribution, the result of diffusion sampling with a $SO(3)$-equivariant denoiser is a $SO(3)$-invariant distribution. In our case, our initial noise distribution is an isotropic multivariate Gaussian, so the conditions are satisfied. Further, for an equivariant denoiser, data augmentation has no effect (Appendix A.3): $l_{\text{aug}}(D) = l_{\text{denoise}}(D)$ for an equivariant $D$.

In Appendix B.1, we show that there is no perfect denoiser under rotational augmentation, implying that $l_{\text{aug}}$ has a non-zero minimum. To summarize, the fundamental issue is that there is an ambiguity with respect to which orientation $\mathbf{R} \circ x$ to denoise to. This inspires the idea of alignment to simply cancel out the effect of any such rotation. However, to analyze the alignment step, we first need to characterize the form of the optimal denoiser $D^*$ which minimizes $l_{\text{aug}}$.

---

[1]We could have also denoted the augmented input to $D$ as $\mathbf{R} \circ x + \sigma\eta$, which is equivalent in expectation due to the isotropy of the Gaussian distribution.

## 3 THE OPTIMAL DENOISER

Having established that there does not exist a perfect denoiser, we can now ask about the optimal denoiser $D^*$ obtaining the minimum possible loss $l_{\text{aug}}(D^*) = \min_D l_{\text{aug}}(D) > 0$. Here, we adapt the derivation performed in Karras et al. (2022).

### 3.1 THE OPTIMAL DENOISER IN THE SINGLE SAMPLE SETTING

We first state the optimal denoiser $D^*$ in the single sample case, where $p_x(x) = \delta(x - x_0)$. Let $D^*(y; x_0, \sigma)$ be the optimal denoiser for $y$ conditional on a particular $x_0$ at a noise level $\sigma$, which we term the *optimal conditional denoiser*. Then:

$$D^*(y; x_0, \sigma) = \frac{\mathbb{E}_{\mathbf{R} \sim u_{\mathbf{R}}}[\mathcal{N}(y; \mathbf{R} \circ x_0, \sigma^2 \mathbb{I}_{N \times 3}) \mathbf{R} \circ x_0]}{\mathbb{E}_{\mathbf{R} \sim u_{\mathbf{R}}}[\mathcal{N}(y; \mathbf{R} \circ x_0, \sigma^2 \mathbb{I}_{N \times 3})]} = \mathbb{E}_{\mathbf{R} \sim p(\mathbf{R} \mid y, x_0, \sigma)}[\mathbf{R} \circ x_0] \qquad (9)$$

as we prove in Appendix B.2. Above:

$$p(\mathbf{R} \mid y, x_0, \sigma) = \frac{\mathcal{N}(y; \mathbf{R} \circ x_0, \sigma^2 \mathbb{I}_{N \times 3}) u_{\mathbf{R}}(\mathbf{R})}{\int_{SO(3)} \mathcal{N}(y; \mathbf{R}' x_0, \sigma^2 \mathbb{I}_{N \times 3}) u_{\mathbf{R}}(\mathbf{R}') d\mathbf{R}'} \qquad (10)$$

Thus, we see that the optimal conditional denoiser essentially corresponds to an expectation over different orientations $\mathbf{R} \circ x$. Importantly, it turns out that the optimal conditional denoiser is $SO(3)$-equivariant:

$$D^*(\mathbf{R} \circ y; x_0, \sigma) = \mathbf{R} \circ D^*(y; x_0, \sigma) \qquad (11)$$

We provide a short proof using the invariance of the Haar measure in Appendix B.5. Further, the optimal conditional denoiser is invariant under rotations $\mathbf{R}_{\text{aug}}$ of the conditioning $x$:

$$D^*(y; \mathbf{R}_{\text{aug}} x, \sigma) = \mathbb{E}_{\mathbf{R} \sim p(\mathbf{R} \mid y, \mathbf{R}_{\text{aug}} \circ x; \sigma)}[\mathbf{R} \circ (\mathbf{R}_{\text{aug}} \circ x)] = D^*(y; x, \sigma) \qquad (12)$$

### 3.2 THE OPTIMAL DENOISER IN THE GENERAL SETTING

In the general setting where $p_x$ is arbitrary, the optimal denoiser $D^*(y; \sigma)$ is an expectation over $x \sim p(x \mid y, \sigma)$ of the optimal conditional denoiser $D^*(y; x, \sigma)$, as we prove in Appendix B.3:

$$D^*(y; \sigma) = \mathbb{E}_{x, \mathbf{R} \sim p(x, \mathbf{R} \mid y, \sigma)}[\mathbf{R} \circ x]$$
$$= \mathbb{E}_{x \sim p(x \mid y, \sigma)} \mathbb{E}_{\mathbf{R} \sim p(\mathbf{R} \mid y, x, \sigma)}[\mathbf{R} \circ x] = \mathbb{E}_{x \sim p(x \mid y, \sigma)}[D^*(y; x, \sigma)] \qquad (13)$$

It follows from the $SO(3)$-equivariance of $D^*(y; x, \sigma)$ that $D^*(y; \sigma)$ is also $SO(3)$-equivariant.

Given that there is an analytic expression of the optimal denoiser, a natural question arises. Instead of minimizing the usual denoising loss (Equation 1), can we instead try to match the optimal denoiser? Indeed, we shall see that these approaches are actually identical.

## 4 MATCHING THE OPTIMAL DENOISER

Here, we motivate why we would want to match the optimal denoiser $D^*$. To do so, we first consider matching to some general estimator $D_{\text{est}}(y; x, \mathbf{R}, \sigma)$ potentially dependent on all the random variables. Given any estimator $D_{\text{est}}(y; x, \mathbf{R}, \sigma)$ we want to match to $D$, we can define a matching loss by:

$$l_{\text{est}}(D; D_{\text{est}}) = \mathbb{E}_{\mathbf{R} \sim u_{\mathbf{R}}} \mathbb{E}_{x \sim p_x} \mathbb{E}_{\eta \sim p_\eta} \mathbb{E}_{\sigma \sim p_\sigma}[\|D(\mathbf{R} \circ (x + \sigma \eta); \sigma) - D_{\text{est}}(\mathbf{R} \circ (x + \sigma \eta); x, \mathbf{R}, \sigma)\|^2]$$
$$= \mathbb{E}_{\mathbf{R} \sim u_{\mathbf{R}}} \mathbb{E}_{x \sim p_x} \mathbb{E}_{\sigma \sim p_\sigma} \mathbb{E}_{y \sim p(y \mid x, \sigma, \mathbf{R})}[\|D(y; \sigma) - D_{\text{est}}(y; x, \mathbf{R}, \sigma)\|^2]$$
$$= \mathbb{E}_{\sigma \sim p_\sigma} \mathbb{E}_{y \sim p(y \mid \sigma)} \mathbb{E}_{x \sim p(x \mid y, \sigma)} \mathbb{E}_{\mathbf{R} \sim p(\mathbf{R} \mid y, x, \sigma)}[\|D(y; \sigma) - D_{\text{est}}(y; x, \mathbf{R}, \sigma)\|^2] \quad (14)$$

after identifying $y \equiv \mathbf{R} \circ (x + \sigma \eta)$. As we prove in Appendix B.7, averaging an estimator $D_{\text{est}}$ over $\mathbf{R} \sim p(\mathbf{R} \mid y, x, \sigma)$ to obtain a new estimator $\mathbb{E}_{\mathbf{R} \sim p(\mathbf{R} \mid y, x, \sigma)}[D_{\text{est}}]$ gives us an equivalent loss from a minimization perspective: $l_{\text{est}}(D; \mathbb{E}_{\mathbf{R} \sim p(\mathbf{R} \mid y, x, \sigma)}[D_{\text{est}}]) = l_{\text{est}}(D; D_{\text{est}}) + C$, where $C$ is a constant that does not depend on $D$. The same reasoning applies to averaging an estimator $D_{\text{est}}$ over $x \sim p(x \mid y, \sigma)$ to obtain a new estimator $\mathbb{E}_{x \sim p(x \mid y, \sigma)}[D_{\text{est}}]$.

Setting $D_{\text{est}}(y; x, \mathbf{R}, \sigma) \equiv \mathbf{R} \circ x$ recovers $l_{\text{aug}}$. Now, for this choice of $D_{\text{est}}$, $\mathbb{E}_{\mathbf{R} \sim p(\mathbf{R} \mid y, x, \sigma)}[D_{\text{est}}]$ corresponds to the *optimal conditional denoiser*. Further, $\mathbb{E}_{x \sim p(x \mid y, \sigma)}\mathbb{E}_{\mathbf{R} \sim p(\mathbf{R} \mid y, x, \sigma)}[D_{\text{est}}]$ corresponds to the *optimal denoiser*. Thus, from the previous arguments, optimal denoiser matching is *equivalent* to the standard denoising loss.

The advantage of averaging, however, comes from a practical perspective. In practice, we can only estimate this loss through sampling which introduces a sampling error for each random variable (ie, $\mathbf{x}, \mathbf{R}$) in the argument. By averaging, we remove the dependence of the argument on the random variable being averaged over and reduce this sampling error.

Performing the averaging over $x$ is tricky, because the expectation over $p(x \mid y, \sigma)$ requires access to $p_x$. Approximating this expectation over a finite set of $x$ can suffer from overfitting (Vastola, 2025; Kadkhodaie et al., 2024; Li et al., 2023). Niedoba et al. (2024) tried to address this by building estimators based on the nearest-neighbor approximation. However, the averaging over $\mathbf{R}$ to give the optimal conditional denoiser is indeed feasible, as we discuss next.

In conclusion, rather than optimizing $l_{\text{aug}}$, we can instead match to the optimal conditional denoiser:

$$l_{\text{est}}(D; D^*(y, x, \sigma)) = l_{\text{match}}(D) = \mathbb{E}_{\sigma \sim p_\sigma}\mathbb{E}_{y \sim p(y|\sigma)}\mathbb{E}_{x \sim p(x \mid y, \sigma)}[\|D(y; \sigma) - D^*(y; x, \sigma)\|^2]. \tag{15}$$

## 5  THE MATRIX FISHER DISTRIBUTION ON $SO(3)$

Here, we connect the optimal conditional denoiser to the matrix Fisher distribution. Recall our expression of $D^*(y; x, \sigma)$ involves an expectation over $p(\mathbf{R} \mid y, x, \sigma)$. Some simple algebraic manipulation (Appendix B.4) shows us that:

$$p(\mathbf{R} \mid y, x, \sigma) \propto \exp\left(-\frac{\|y - \mathbf{R} \circ x\|^2}{2\sigma^2}\right) \propto \exp\left(\text{Tr}\left[\frac{y^\top x}{2\sigma^2}\mathbf{R}\right]\right) \tag{16}$$

In particular, the distribution $p(\mathbf{R} \mid y, x, \sigma)$ belongs to a well-studied family of distributions called the matrix Fisher distribution over $SO(3)$ (Mohlin et al., 2020; Lee, 2018). This distribution is parametrized by a $3 \times 3$ matrix $F$:

$$\text{MF}(\mathbf{R}; F) = \frac{\exp \text{Tr}[F^\top \mathbf{R}]}{Z(F)} \tag{17}$$

where $Z(F)$ is the partition function, ensuring that the distribution is normalized over $SO(3)$. Thus, conditional on $y, x$ and $\sigma$, $\mathbf{R}$ is distributed according to a matrix Fisher distribution with $F = \frac{y^\top x}{\sigma^2} \in \mathbb{R}^{3 \times 3}$. We can write:

$$D^*(y; x, \sigma) = \mathbb{E}_{\mathbf{R} \sim \text{MF}\left(\mathbf{R}; \frac{y^\top x}{\sigma^2}\right)}[\mathbf{R} \circ x] = \mathbb{E}_{\mathbf{R} \sim \text{MF}\left(\mathbf{R}; \frac{y^\top x}{\sigma^2}\right)}[\mathbf{R}] \circ x. \tag{18}$$

Therefore computing the optimal denoiser involves computing the first moment of a matrix Fisher distribution.

## 6  ROTATIONAL ALIGNMENT

It turns out rotation alignment gives a very good approximation for the first moment. Assuming $x, y$ have no rotational self symmetries (which is generically the case), then $\text{MF}\left(\mathbf{R}; \frac{y^\top x}{\sigma^2}\right)$ is a unimodal distribution. In fact, for small $\sigma$ this distribution becomes very sharply peaked, so the mode becomes a good approximation of the first moment.

We can see the mode of the distribution satisfies:

$$\mathbf{R}_{\text{mode}} = \underset{\mathbf{R} \in SO(3)}{\text{argmax}} \, \text{MF}\left(\mathbf{R}; \frac{y^\top x}{\sigma^2}\right) = \underset{\mathbf{R} \in SO(3)}{\text{argmax}} \, \text{Tr}\left[\frac{y^\top x}{2\sigma^2}\mathbf{R}\right] = \underset{\mathbf{R} \in SO(3)}{\text{argmin}} \, -\text{Tr}[y^\top x\mathbf{R}]$$

$$= \underset{\mathbf{R} \in SO(3)}{\text{argmin}} \, \frac{1}{2}(\text{Tr}[y^\top y] - 2\text{Tr}[y^\top x\mathbf{R}] + \text{Tr}[\mathbf{R}^\top x^\top x\mathbf{R}]) = \underset{\mathbf{R} \in SO(3)}{\text{argmin}} \, \|y - \mathbf{R} \circ x\|^2 \tag{19}$$

$$= \mathbf{R}^*(y, x) \tag{20}$$

where $\mathbf{R}^*(y, x)$ is the optimal rotation for aligning $x$ to $y$, exactly the rotation matrix returned by the Kabsch and Proscrutes alignment algorithms.

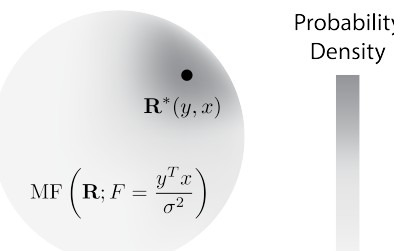

Figure 2: A depiction of the unimodal $\mathrm{MF}(\mathbf{R}; F)$ over $SO(3)$, highlighting the mode $\mathbf{R}^*(y, x)$. As $\sigma$ decreases, the distribution becomes more peaked around $\mathbf{R}^*(y, x)$.

This marks our first insight: alignment allows us to approximate $\mathbb{E}_{\mathbf{R}}[\mathbf{R}] \approx \mathbf{R}^*(y, x)$ in the optimal single sample denoiser (Equation 9) so

$$D^*(y; x, \sigma) = \mathbb{E}_{\mathbf{R}}[\mathbf{R}] \circ x \approx \mathbf{R}^*(y, x)x.$$

Substituting $\mathbf{R}^*(y, x)x$ for $D_{\mathrm{est}}$ in Equation 14, we obtain

$$l_{\text{align-aug}}(D) = \mathbb{E}_{\mathbf{R} \sim u_{\mathbf{R}}} \mathbb{E}_{x \sim p_x} \mathbb{E}_{\eta \sim p_\eta} \mathbb{E}_{\sigma \sim p_\sigma} [\|D(\mathbf{R} \circ (x + \sigma\eta); \sigma) - \mathbf{R}^*(y, x) \circ x\|^2] \qquad (21)$$

exactly the loss used for rotation alignment!

Can better estimators for $D^*$ be constructed by better approximations of $\mathbb{E}_{\mathbf{R}}[\mathbf{R}]$? Yes! In Section 7, we show that there exist better approximators of $\mathbb{E}_{\mathbf{R}}[\mathbf{R}]$ with no additional asymptotic runtime cost over alignment.

# 7 APPROXIMATING THE OPTIMAL DENOISER VIA AN ASYMPTOTIC EXPANSION OF THE MATRIX-FISHER DISTRIBUTION

Here, we derive additional correction terms to the moment $\mathbb{E}_{\mathbf{R}}[\mathbf{R}]$ in the limit of $\sigma \to 0$. These correction terms can be implemented with no additional cost.

## 7.1 FOR A GENERAL MATRIX FISHER DISTRIBUTION

We provide a high level summary of the method used to derive additional correction terms. More detail can be found in Appendix C.

We note that the partition function $Z(F) = \int_{SO(3)} \exp(\mathrm{Tr}[F^\top \mathbf{R}]) d\mathbf{R}$ gives all the information we need. In particular, taking derivatives of $Z(F)$ allows us to calculate any necessary moments. For example,

$$\mathbb{E}_{\mathbf{R} \sim \mathrm{MF}(\mathbf{R}; F)}[\mathbf{R}] = \frac{\int_{SO(3)} \mathbf{R} \exp(\mathrm{Tr}[F^\top \mathbf{R}]) d\mathbf{R}}{Z(F)} = \frac{\frac{d}{dF} Z(F)}{Z(F)} = \frac{d}{dF} \ln Z(F) \qquad (22)$$

using the trace derivative identity: $\frac{d}{dF} \mathrm{Tr}[F^\top \mathbf{R}] = \mathbf{R}$. To control peakedness, we replace $F$ with $\lambda F$ so that as $\lambda \to \infty$, the distribution becomes a delta function centered at the mode.

Next, recall that in Kabsch alignment we use SVD to decompose where $U, V \in SO(3)$ and $S = \mathrm{diag}[s_1, s_2, s_3]$ where $s_1 \geq s_2 \geq |s_3|$. It turns out that $Z(\lambda F) = Z(\lambda S)$ so we restrict to only expanding $Z$ for diagonal $S$. To approximate $Z(\lambda S)$, we use Laplace's method.

First, we choose to use the exponential map parameterization $\mathbf{R}(\theta_x, \theta_y, \theta_z) = \exp(\theta_x R_x + \theta_y R_y + \theta_z R_z)$. Next, we Taylor expand the argument $\mathrm{Tr}[S^\top \mathbf{R}(\boldsymbol{\theta})]$ around $\boldsymbol{\theta} = 0$. It is not hard to check that $\mathbb{I}$ maximizes this so there is no first order term. Hence taking up to second order terms, we obtain some $\exp(\lambda A_0(S) + \lambda \boldsymbol{\theta}^\top A_2(S) \boldsymbol{\theta})$ which can be interpreted as a Gaussian because $A_2$ must be negative

definite since $\mathbb{I}$ is the maximum. In some sense this captures the peak of the distribution and we can write

$$\exp(\mathrm{Tr}[\lambda S^\top \mathbf{R}(\boldsymbol{\theta})]) = \exp(\lambda A_0(S) + \lambda \theta^\top A_2(S)\theta)B(\boldsymbol{\theta}, S, \lambda)$$

Next, we can perform the usual Taylor expansion of $B(\boldsymbol{\theta}, S, \lambda)\mu(\boldsymbol{\theta})$, the remaining terms in the integral where $\mu(\boldsymbol{\theta})$ is the corresponding Haar measure. We also do this around $\boldsymbol{\theta} = 0$ because only the neighborhood around the peak matters. Finally, we note that as $\lambda \to \infty$, the width of the peak described by $\exp(\lambda A_0(S) + \lambda \theta^\top A_2(S)\theta)$ decreases. Hence, replacing the domain $\{|\boldsymbol{\theta}| < \pi\}$ with the larger domain $\{\boldsymbol{\theta} \in \mathbb{R}^3\}$ not gives us a good approximation for each of the expanded terms, but also gives us Gaussian integrals which are analytically evaluable.

Finally, we obtain an expression of the form:

$$Z(\lambda S) = N(S, \lambda)\left(1 + L_1(S)\frac{1}{\lambda} + L_2(S)\frac{1}{\lambda^2} + L_3(S)\frac{1}{\lambda^3} + \dots\right) \tag{23}$$

where $N(S, \lambda)$ is a normalization term. The corresponding expected rotation can be computed as:

$$\mathbb{E}_{\mathbf{R}\sim\mathrm{MF}(\mathbf{R};\lambda S)}[\mathbf{R}] = \frac{1}{\lambda}\,\mathrm{diag}\left[\frac{\partial \ln Z(\lambda S)}{\partial s_1}, \frac{\partial \ln Z(\lambda S)}{\partial s_1}, \frac{\partial \ln Z(\lambda S)}{\partial s_3}\right]$$

$$= \mathbb{I} + C_1(S)\frac{1}{\lambda} + C_2(S)\frac{1}{\lambda^2} + \dots \tag{24}$$

For an arbitrary $F' = USV^\top$, we would then have:

$$\mathbb{E}_{\mathbf{R}\sim\mathrm{MF}(\mathbf{R};\lambda F')}[\mathbf{R}] = U\mathbb{E}_{\mathbf{R}\sim\mathrm{MF}(\mathbf{R};\lambda S)}[\mathbf{R}]V^\top. \tag{25}$$

## 7.2 FOR THE SPECIFIC $F = \frac{y^\top x}{\sigma^2}$

We are specifically interested in the case where $F = \frac{y^\top x}{\sigma^2}$, as $p(\mathbf{R} \mid y, x, \sigma) = \mathrm{MF}(\mathbf{R}; \frac{y^\top x}{\sigma^2})$. From the Kabsch algorithm, we have that $\mathbf{R}^*(y, x) = UV^\top$ where $U, S, V^\top = \mathrm{SVD}(y^\top x) = \mathrm{SVD}(F')$ with $\det(U) = \det(V) = 1$ and $S = \mathrm{diag}[s_1, s_2, s_3]$ where $s_1 \geq s_2 \geq |s_3|$.

Following the procedure outlined in Section 7, we used Mathematica (Inc.) to find the coefficients in Equation 24 explicitly:

$$C_1(S) = -\frac{1}{2}\,\mathrm{diag}\left[\frac{1}{s_1 + s_2} + \frac{1}{s_1 + s_3}, \frac{1}{s_2 + s_1} + \frac{1}{s_2 + s_3}, \frac{1}{s_3 + s_1} + \frac{1}{s_3 + s_2}\right] \tag{26}$$

$$C_2(S) = -\frac{1}{8}\,\mathrm{diag}\left[\frac{1}{(s_1 + s_2)^2} + \frac{1}{(s_1 + s_3)^2}, \frac{1}{(s_2 + s_1)^2} + \frac{1}{(s_2 + s_3)^2}, \frac{1}{(s_3 + s_1)^2} + \frac{1}{(s_3 + s_2)^2}\right] \tag{27}$$

These represent the first-order and second-order correction terms respectively. This approximation is exact in the limit $\sigma \to 0$.

Equation 24 allows us to approximate the optimal conditional denoiser as:

$$D^*(y; x) = \mathbb{E}_{\mathbf{R}\sim p(\mathbf{R} \mid y, x, \sigma)}[\mathbf{R} \circ x] = \mathbb{E}_{\mathbf{R}\sim p(\mathbf{R} \mid y, x, \sigma)}[\mathbf{R}] \circ x \tag{28}$$

$$= (\mathbf{R}^*(y, x) + \sigma^2 B_1(y, x) + \sigma^4 B_2(y, x)) \circ x + \mathcal{O}(\sigma^5). \tag{29}$$

where $B_1(y, x) = UC_1(S)V^\top$ and $B_2(y, x) = UC_2(S)V^\top$.

Thus, alignment corresponds to the *zeroth-order* (in $\sigma$) approximation of $D^*(y; x, \sigma)$. From Equation 29, we define the successive *first-order* and *second-order* approximations to the optimal denoiser:

$$D_0^*(y; x, \sigma) = \mathbf{R}^*(y, x) \circ x \tag{30}$$

$$D_1^*(y; x, \sigma) = (\mathbf{R}^*(y, x) + \sigma^2 B_1(y, x)) \circ x \tag{31}$$

$$D_2^*(y; x, \sigma) = (\mathbf{R}^*(y, x) + \sigma^2 B_1(y, x) + \sigma^4 B_2(y, x)) \circ x \tag{32}$$

Our improved estimators $D_1^*$ and $D_2^*$ have reduced bias relative to the usual alignment-based $D_0^*$. Importantly, these improved estimators can be constructed at no additional computational cost to the standard Kabsch alignment, since $U$ and $V$ have already been computed. They simply correspond to adjusting the rotation $\mathbf{R}^*$ before multiplying with $x$.

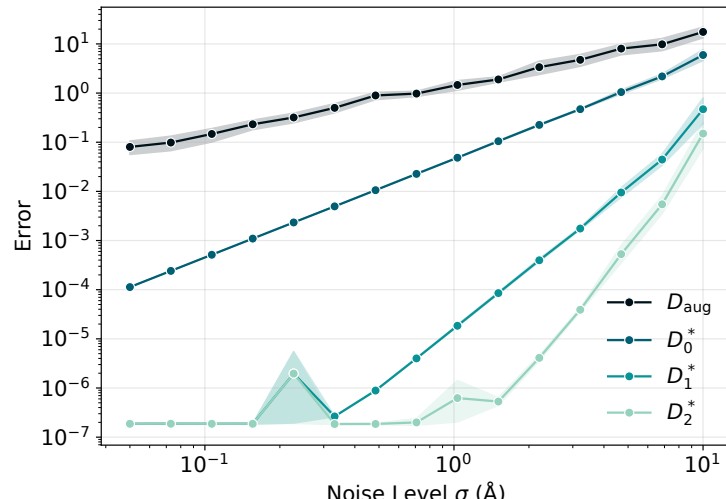

Figure 3: Mean-squared error relative to the optimal denoiser $D^*(y; x)$ as a function of $\sigma$. $x$ here is a randomly chosen conformation of the AEQN tetrapeptide from the TIMEWARP 4AA-LARGE dataset, and $y$ is sampled as $x + \sigma\eta$. For larger $\sigma$, we see that the error rates drop significantly using higher order correction terms. For smaller $\sigma$, we quickly reach the regime where error is dominated by numerical precision rather than approximation error when using higher order correction terms.

## 8 NUMERICAL ERROR IN APPROXIMATIONS OF THE OPTIMAL DENOISER

In Figure 3, we compute error in the zeroth-order $D_0^*$, first-order $D_1^*$, and second-order $D_2^*$ approximations compared to an estimate of $D^*(y; x)$ obtained by numerically integrating the expectation in Equation 28 in Mathematica, which automatically adjusts the resolution of the grid for the numerical quadrature on $SO(3)$.

In the next section, we experiment with the practical utility of these estimators.

## 9 RESULTS IN PRACTICE

We train a simple 2-layer MLP (with 2.3M parameters) on 3D configurations of the tetrapeptide AEQN as obtained from the TIMEWARP 4AA-LARGE dataset (Klein et al., 2023a), utilizing the codebase of the JAMUN (Daigavane et al., 2025) model. We train this model using the losses corresponding to the estimators $D_{\text{aug}}, D_0^*, D_1^*$ and $D_2^*$ discussed above, at 4 different noise levels $\sigma = 0.5\,\text{Å}$, $1.0\,\text{Å}$, $5\,\text{Å}$ and $10\,\text{Å}$. This captures most of the noise levels usually used to train diffusion models on data of this size (Wohlwend et al., 2024). We perform two experiments which allow us to measure the impact of the estimators in learning 1. the optimal denoiser and 2. the optimal conditional denoiser.

1. We sample $x$ from all 50000 frames of a molecular dynamics simulation for the AEQN peptide, as obtained from Klein et al. (2023a).

2. We fix $x$ as the first frame of the same molecular dynamics simulation.

The results are shown in Figure 4 and Figure 5, where we report the RMSD (root mean square deviation) to the ground truth $x$. In Appendix D, we also show plots of the aligned RMSD for the same training runs.

We see that at the largest noise level, the second-order correction tends to diverge. At the lowest noise levels, the magnitude of the correction is not significant, and all estimators perform similarly. We see that in practice, the zeroth-order approximation $D_0^*$ is usually good enough at the important noise levels; in particular, it seems like the model is unable to take advantage of the variance reduction from the higher-order corrections. We hypothesize that this may be due to the fact that the variance of the

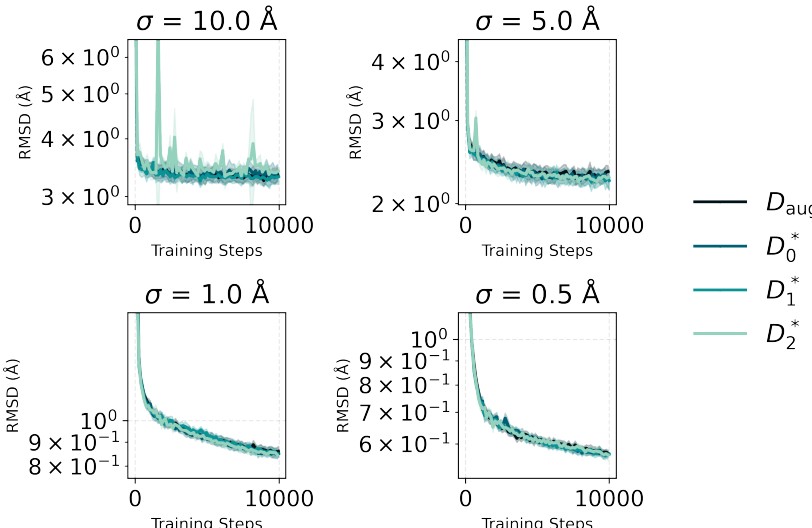

Figure 4: Training progress for the MLP, as measured by RMSD to ground-truth $x$ (sampled from all frames), when trained using $D_{\text{aug}}$, $D_0^*$, $D_1^*$ and $D_2^*$.

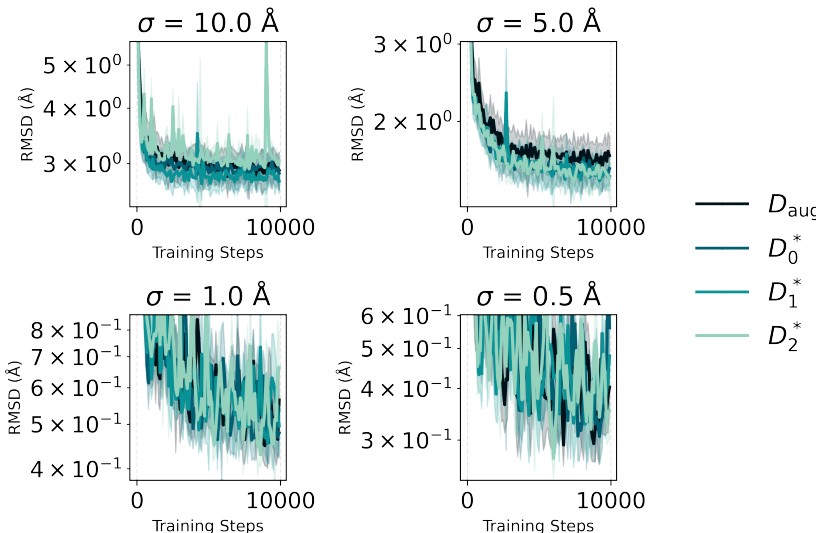

Figure 5: Training progress for the MLP, as measured by RMSD to ground-truth $x$ (fixed as the first frame), when trained using $D_{\text{aug}}$, $D_0^*$, $D_1^*$ and $D_2^*$.

gradients over the multiple $x$ is much greater than the variance of the gradients due to the rotational symmetries. These results are preliminary, but they suggest that alignment itself may not be super critical to learn a good denoiser.

## 10   ETHICS STATEMENT

We have read and agree to comply to the ICLR Code of Ethics. Our results are primarily theoretical and specific to the field of computational biology; we do not anticipate any negative impacts on society from this research.

## 11   REPRODUCIBILITY STATEMENT

We have provided the Mathematica notebooks, our training code, and data (obtained from the TIMEWARP dataset, with the authors' permission) to reproduce all of our results as supplementary material.

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

## A  PROOFS

### A.1  EQUIVARIANT DENOISERS SAMPLE INVARIANT DISTRIBUTIONS

The proof is similar to that of Proposition 1 in Xu et al. (2022).

Suppose that $p_y(; \sigma_{i+1})$ is a $SO(3)$-invariant distribution:

$$p_y(y_{i+1}; \sigma_{i+1}) = p_y(\mathbf{R} \circ y_{i+1}; \sigma_{i+1}) \tag{33}$$

Using the DDIM update rule (Equation 2) at step $(i + 1)$, we see that the next distribution $p_y(; \sigma_i)$ is also $SO(3)$-invariant. Under an arbitrary rotation $\mathbf{R}$:

$$\text{DDIM}_{i+1}(\mathbf{R} \circ y_{i+1}) = \mathbf{R} \circ y_{i+1} + \left(1 - \frac{\sigma_i}{\sigma_{i+1}}\right)(D(\mathbf{R} \circ y_{i+1}, \sigma_{i+1}) - \mathbf{R} \circ y_{i+1}) \tag{34}$$

$$= \mathbf{R} \circ y_{i+1} + \left(1 - \frac{\sigma_i}{\sigma_{i+1}}\right)(\mathbf{R} \circ D(y_{i+1}, \sigma_{i+1}) - \mathbf{R} \circ y_{i+1}) \tag{35}$$

$$= \mathbf{R} \circ \left(y_{i+1} + \left(1 - \frac{\sigma_i}{\sigma_{i+1}}\right)(D(y_{i+1}, \sigma_{i+1}) - y_{i+1})\right) \tag{36}$$

$$= \mathbf{R} \circ y_{i+1} \tag{37}$$

$$= \mathbf{R} \circ \text{DDIM}_{i+1}(y_{i+1}) \tag{38}$$

We used the $SO(3)$-equivariance of the denoiser above: $D(\mathbf{R} \circ y_{i+1}, \sigma_{i+1}) = \mathbf{R} \circ D(y_{i+1}, \sigma_{i+1})$ for all $y_{i+1}$.

Hence:

$$p_y(y_i; \sigma_i) = \int p(y_i | y_{i+1}) p_y(y_{i+1}; \sigma_{i+1}) dy_{i+1} \tag{39}$$

$$= \int \delta(y_i - \text{DDIM}_{i+1}(y_{i+1})) p_y(y_{i+1}; \sigma_{i+1}) dy_{i+1} \tag{40}$$

$$= \int \delta(y_i - \text{DDIM}_{i+1}(y_{i+1})) p_y(\mathbf{R} \circ y_{i+1}; \sigma_{i+1}) dy_{i+1} \tag{41}$$

$$= \int \delta(\mathbf{R} \circ y_i - \mathbf{R} \circ \text{DDIM}_{i+1}(y_{i+1})) p_y(\mathbf{R} \circ y_{i+1}; \sigma_{i+1}) dy_{i+1} \tag{42}$$

$$= \int \delta(\mathbf{R} \circ y_i - \text{DDIM}_{i+1}(\mathbf{R} \circ y_{i+1})) p_y(\mathbf{R} \circ y_{i+1}; \sigma_{i+1}) dy_{i+1} \tag{43}$$

$$= \int \delta(\mathbf{R} \circ y_i - \text{DDIM}_{i+1}(y'_{i+1})) p_y(y'_{i+1}; \sigma_{i+1}) dy'_{i+1} \tag{44}$$

$$= p_y(\mathbf{R} \circ y_i; \sigma_i) \tag{45}$$

using the change of variables $y'_{i+1} \equiv \mathbf{R} \circ y_{i+1}$ which does not induce any change of measure. Hence,

$$p_y(; \sigma_i)$$

is a $SO(3)$-invariant distribution. The initial distribution is $\mathcal{N}(0, \sigma_M^2 \mathbb{I}_d)$, which is also $SO(3)$-invariant distribution due to the isotropy of the multivariate Gaussian. We can conclude that at each noise level $\sigma_i$ in the diffusion process, the $p_y(; \sigma_i)$ is $SO(3)$-invariant.

The same proof can be generalized to stochastic samplers such as DDSM (Yang et al., 2024).

## A.2 ROTATIONALLY AUGMENTED DISTRIBUTIONS ARE INVARIANT

For any arbitrary rotation $\mathbf{R}$:

$$\text{Aug}[p_x](\mathbf{R} \circ x) = \int_{SO(3)} p_x(\mathbf{R}'^{-1}\mathbf{R} \circ x)u_{\mathbf{R}}(\mathbf{R}')\text{d}\mathbf{R}' \tag{46}$$

$$= \int_{SO(3)} p_x((\mathbf{R}^{-1}\mathbf{R}')^{-1} \circ x)u_{\mathbf{R}}(\mathbf{R}(\mathbf{R}^{-1}\mathbf{R}'))\text{d}(\mathbf{R}\mathbf{R}^{-1}\mathbf{R}') \tag{47}$$

$$= \int_{SO(3)} p_x(\mathbf{R}''^{-1} \circ x)u_{\mathbf{R}}(\mathbf{R}\mathbf{R}'')\text{d}(\mathbf{R}\mathbf{R}'') \tag{48}$$

$$= \int_{SO(3)} p_x(\mathbf{R}''^{-1} \circ x)u_{\mathbf{R}}(\mathbf{R}'')\text{d}\mathbf{R}'' \tag{49}$$

$$= \text{Aug}[p_x](x). \tag{50}$$

## A.3 ROTATIONAL AUGMENTATION DOES NOT AFFECT EQUIVARIANT DENOISERS

Using the equivariance of $D$ and the fact that for any rotation $\mathbf{R}$, we have $\mathbf{R}^T\mathbf{R} = \mathbb{I}_{3\times3}$:

$$\|D(\mathbf{R}(x+\eta)) - \mathbf{R}x\|^2 = \|\mathbf{R}D(x+\eta) - \mathbf{R}x\|^2 \tag{51}$$

$$= \|\mathbf{R}(D(x+\eta) - x)\|^2 \tag{52}$$

$$= (\mathbf{R}(D(x+\eta) - x))^T\mathbf{R}(D(x+\eta) - x) \tag{53}$$

$$= (D(x+\eta) - x))^T\mathbf{R}^T\mathbf{R}(D(x+\eta) - x) \tag{54}$$

$$= (D(x+\eta) - x))^T(D(x+\eta) - x) \tag{55}$$

$$= \|(D(x+\eta) - x)\|^2 \tag{56}$$

Hence, the loss $l_{\text{no-aug}}(D)$ is invariant under rotations $\mathbf{R}$. Thus, if $D$ is equivariant:

$$l_{\text{no-aug}}(D) = \mathbb{E}_{\mathbf{R}\sim u_{\mathbf{R}}}\mathbb{E}_{x\sim p_x}\mathbb{E}_{\eta\sim p_\eta}\|D(\mathbf{R}(x+\eta)) - \mathbf{R}x\|^2 = l_{\text{aug}}(D) \tag{57}$$

# B  NO PERFECT DENOISER EXISTS WITH ROTATIONAL AUGMENTATION

Suppose we only had one sample $x_0$ where $\|x_0\| > 0$. Hence, our distribution is a delta function $p_x(x) = \delta(x - x_0)$. Fix a noise level $\sigma > 0$. Here, we argue why a perfect denoiser cannot exist[2].

In this setting, assume that there exists a perfect denoiser $D_{\text{perf}}$:

$$D_{\text{perf}}(\mathbf{R} \circ (x_0 + \sigma\eta)) = \mathbf{R} \circ x_0 \tag{58}$$

for all rotations $\mathbf{R}$ and noise $\eta$. Such a $D_{\text{perf}}$ obtains zero loss: $l_{\text{aug}}(D_{\text{perf}}) = 0$.

We show that such a $D_{\text{perf}}$ cannot exist, by contradiction. Set $\mathbf{R} = \mathbb{I}$ in Equation 58, to see that $D_{\text{perf}}(x_0 + \sigma\eta) = x_0$ for all instantiations of $\eta$. Fix some noise $\eta_0$ arbitrarily. Now, consider any non-identity rotation $\mathbf{R}$. Let $\eta_{\mathbf{R}}$ be such that:

$$\mathbf{R} \circ (x_0 + \sigma\eta_{\mathbf{R}}) = x_0 + \sigma\eta_0 \quad \implies \quad \eta_{\mathbf{R}} = \frac{\mathbf{R}^\top \circ (x_0 + \sigma\eta_0) - x_0}{\sigma} \tag{59}$$

where we used the fact that $\mathbf{R}$ is orthonormal. Now, setting $\eta = \eta_{\mathbf{R}}$ in Equation 58:

$$D_{\text{perf}}(\mathbf{R} \circ (x_0 + \sigma\eta_{\mathbf{R}})) = \mathbf{R} \circ x_0. \tag{60}$$

But from the definition of $\eta_{\mathbf{R}}$, we have:

$$D_{\text{perf}}(\mathbf{R} \circ (x_0 + \sigma\eta_{\mathbf{R}})) = D_{\text{perf}}(x_0 + \sigma\eta_0) = x_0. \tag{61}$$

As $\mathbf{R}$ is not the identity, we have a contradiction. Thus, $D_{\text{perf}}$ cannot exist.

---

[2]This argument can be made rigorous to allow for exceptions of zero measure. We do this in Appendix B.1, but ignore such exceptions for clarity here.

### B.1 NO PERFECT DENOISER EXISTS WITH ROTATIONAL AUGMENTATION

We can formalize the argument of Appendix B to include exceptions of measure 0.

Assume that there exists a perfect denoiser $D_{\text{perf}}$:

$$D_{\text{perf}}(\mathbf{R} \circ (x_0 + \sigma\eta)) = \mathbf{R} \circ x_0 \tag{62}$$

for all rotations $\mathbf{R}$ and noise $\eta$, except on a set $S$ of measure 0 over $SO(3) \times \mathbb{R}^{N \times 3}$. Now, define the family of sets:

$$R(y) = \{(\mathbf{R}, \eta) : \mathbf{R} \circ (x_0 + \sigma\eta) = y\} \tag{63}$$

for each $y \in \mathbb{R}^{N \times 3}$. It is easy to verify that $\{R(y)\}_{y \in \mathbb{R}^{N \times 3}}$ is a partition of the space $SO(3) \times \mathbb{R}^{N \times 3}$. Further, each $R(y)$ is diffeomorphic to $SO(3)$, since we can always find a $\eta$ for every $\mathbf{R}$ to satisfy $\mathbf{R} \circ (x_0 + \sigma\eta) = y$. Thus, we have $\mu_{SO(3)}(R(y)) = 1$ where $\mu_{SO(3)}$ is the Haar measure over $SO(3)$.

Thus, we can measure any measurable set $A$ by integrating the measure of its intersections with each $R(y)$. Formally, by the co-area formula, where $\mu$ represents the product measure over $SO(3) \times \mathbb{R}^{N \times 3}$:

$$\mu(A) = \int_{y \in \mathbb{R}^{N \times 3}} \mu_{SO(3)}(A \cap R(y)) J(y) dy \tag{64}$$

where $J(y) > 0$ is the Jacobian factor. (Essentially, this is the 'change of variables' formula.) Now, applying this to $S$ with $\mu(S) = 0$, we see that:

$$\int_{y \in \mathbb{R}^{N \times 3}} \mu_{SO(3)}(S \cap R(y)) J(y) dy = 0 \tag{65}$$

$$\implies \mu_{SO(3)}(S \cap R(y)) = 0 \text{ for almost every } y \in \mathbb{R}^{N \times 3} \tag{66}$$

Pick any such $y$ . Then,

$$\mu_{SO(3)}(R(y) - S) = \mu_{SO(3)}(R(y)) - \mu_{SO(3)}(S \cap R(y)) = 1 - 0 = 1. \tag{67}$$

Thus, for this particular $y$, $\mu_{SO(3)}(R(y) - S) = 1 > 0 \implies R(y) - S$ is uncountable, and has infinitely many points. In particular, there exist atleast two different points $(\mathbf{R}_1, \eta_1)$ and $(\mathbf{R}_2, \eta_2)$ in $R(y) - S$ such that $\mathbf{R}_1 \neq \mathbf{R}_2$. Now, for each of these two points:

$$\mathbf{R}_1 \circ (x_0 + \sigma\eta_1) = y = \mathbf{R}_2 \circ (x_0 + \sigma\eta_2). \tag{68}$$

But, since $(\mathbf{R}_1, \eta_1)$ and $(\mathbf{R}_2, \eta_2)$ in $R(y) - S$ (and hence, not in $S$), we must have perfect denoising for the corresponding inputs:

$$\mathbf{R}_1 \circ x_0 = D_{\text{perf}}(\mathbf{R}_1 \circ (x_0 + \sigma\eta_1)) = D_{\text{perf}}(y) = D_{\text{perf}}(\mathbf{R}_2 \circ (x_0 + \sigma\eta_2)) = \mathbf{R}_2 \circ x_0 \tag{69}$$

which is a contradiction as $\|x_0\| > 0$ and $\mathbf{R}_1 \neq \mathbf{R}_2$. Thus, perfect denoising is impossible.

### B.2 THE OPTIMAL DENOISER IN THE SINGLE SAMPLE SETTING

In the single sample case where $p_x(x) = \delta(x - x_0)$, using the isotropy of the Gaussian $p_\eta$:

$$l_{\text{aug}}(D) = \mathbb{E}_{\mathbf{R} \sim u_{\mathbf{R}}} \mathbb{E}_{\eta \sim \mathcal{N}(0, \sigma^2 \mathbb{I}_{N \times 3})} \|D(\mathbf{R}(x_0 + \eta)) - \mathbf{R}x_0\|^2 \tag{70}$$

$$= \mathbb{E}_{\mathbf{R} \sim u_{\mathbf{R}}} \mathbb{E}_{\eta \sim \mathcal{N}(0, \sigma^2 \mathbb{I}_{N \times 3})} \|D(\mathbf{R}x_0 + \eta) - \mathbf{R}x_0\|^2 \tag{71}$$

$$= \mathbb{E}_{\mathbf{R} \sim u_{\mathbf{R}}} \mathbb{E}_{y \sim \mathcal{N}(\mathbf{R}x_0, \sigma^2 \mathbb{I}_{N \times 3})} \|D(y) - \mathbf{R}x_0\|^2 \tag{72}$$

$$= \int_{SO(3)} \left( \int_{\mathbb{R}^{N \times 3}} \|D(y) - \mathbf{R}x_0\|^2 \mathcal{N}(y; \mathbf{R}x_0, \sigma^2 \mathbb{I}_{N \times 3}) dy \right) u_{\mathbf{R}}(\mathbf{R}) d\mathbf{R} \tag{73}$$

$$= \int_{\mathbb{R}^{N \times 3}} \left( \int_{SO(3)} \|D(y) - \mathbf{R}x_0\|^2 \mathcal{N}(y; \mathbf{R}x_0, \sigma^2 \mathbb{I}_{N \times 3}) u_{\mathbf{R}}(\mathbf{R}) d\mathbf{R} \right) dy \tag{74}$$

$$= \int_{\mathbb{R}^{N \times 3}} l_{\text{aug}}(D; y, \sigma) dy \tag{75}$$

where we define:

$$l_{\text{aug}}(D; y, \sigma) = \int_{SO(3)} \|D(y) - \mathbf{R}x_0\|^2 \mathcal{N}(y; \mathbf{R}x_0, \sigma^2 \mathbb{I}_{N \times 3}) u_{\mathbf{R}}(\mathbf{R}) d\mathbf{R} \tag{76}$$

$$= \mathbb{E}_{\mathbf{R} \sim u_{\mathbf{R}}} \|D(y) - \mathbf{R}x_0\|^2 \mathcal{N}(y; \mathbf{R}x_0, \sigma^2 \mathbb{I}_{N \times 3}) \tag{77}$$

Note that $l_{\text{aug}}(D; y)$ is non-negative for all $y$. Thus, the optimal denoiser $D^*$ should minimize $l_{\text{aug}}(D; y)$ for each possible $y$. Taking the gradient of $l_{\text{aug}}(D; y)$ with respect to $D(y)$ and setting it to 0:

$$\nabla_{D(y)} l_{\text{aug}}(D; y, \sigma) = 0 \tag{78}$$

$$\implies \mathbb{E}_{\mathbf{R} \sim u_{\mathbf{R}}} \left[ 2(D^*(y) - \mathbf{R}x_0) \mathcal{N}(y; \mathbf{R}x_0, \sigma^2 \mathbb{I}_{N \times 3}) \right] = 0 \tag{79}$$

$$\implies D^*(y) = \frac{\mathbb{E}_{\mathbf{R} \sim u_{\mathbf{R}}}[\mathbf{R}x_0 \, \mathcal{N}(y; \mathbf{R}x_0, \sigma^2 \mathbb{I}_{N \times 3})]}{\mathbb{E}_{\mathbf{R} \sim u_{\mathbf{R}}}[\mathcal{N}(y; \mathbf{R}x_0, \sigma^2 \mathbb{I}_{N \times 3})]} \tag{80}$$

This is the optimal denoiser in the single sample setting! We can rewrite this a bit:

$$D^*(y) = \frac{\int_{SO(3)} \mathbf{R}x_0 \mathcal{N}(y; \mathbf{R}x_0, \sigma^2 \mathbb{I}_{N \times 3}) u_{\mathbf{R}}(\mathbf{R}) d\mathbf{R}}{\int_{SO(3)} \mathcal{N}(y; \mathbf{R}x_0, \sigma^2 \mathbb{I}_{N \times 3}) u_{\mathbf{R}}(\mathbf{R}) d\mathbf{R}} \tag{81}$$

$$= \frac{\int_{SO(3)} \mathbf{R}x_0 \mathcal{N}(y; \mathbf{R}x_0, \sigma^2 \mathbb{I}_{N \times 3}) u_{\mathbf{R}}(\mathbf{R}) d\mathbf{R}}{\int_{SO(3)} \mathcal{N}(y; \mathbf{R}'x_0, \sigma^2 \mathbb{I}_{N \times 3}) u_{\mathbf{R}}(\mathbf{R}') d\mathbf{R}'} \tag{82}$$

$$= \int_{SO(3)} \mathbf{R}x_0 \frac{\mathcal{N}(y; \mathbf{R}x_0, \sigma^2 \mathbb{I}_{N \times 3}) u_{\mathbf{R}}(\mathbf{R}) d\mathbf{R}}{\int_{SO(3)} \mathcal{N}(y; \mathbf{R}'x_0, \sigma^2 \mathbb{I}_{N \times 3}) u_{\mathbf{R}}(\mathbf{R}') d\mathbf{R}'} \tag{83}$$

$$= \int_{SO(3)} \mathbf{R}x_0 \, p(\mathbf{R} \mid y, x_0) d\mathbf{R} \tag{84}$$

$$= \mathbb{E}_{\mathbf{R} \sim p(\mathbf{R} \mid y, x_0)}[\mathbf{R}x_0] \tag{85}$$

as:

$$p(\mathbf{R} \mid y, x_0) = \frac{\mathcal{N}(y; \mathbf{R}x_0, \sigma^2 \mathbb{I}_{N \times 3}) u_{\mathbf{R}}(\mathbf{R})}{\int_{SO(3)} \mathcal{N}(y; \mathbf{R}'x_0, \sigma^2 \mathbb{I}_{N \times 3}) u_{\mathbf{R}}(\mathbf{R}') d\mathbf{R}'} \tag{86}$$

is the distribution over rotations $\mathbf{R}$ conditional on $y$, because:

$$p(y \mid \mathbf{R}, x_0) = \mathcal{N}(y; \mathbf{R}x_0, \sigma^2 \mathbb{I}_{N \times 3}) \tag{87}$$

and Bayes' rule:

$$p(\mathbf{R} \mid y, x_0) = \frac{p(y, \mathbf{R} \mid x_0)}{p(y \mid x_0)} = \frac{p(y \mid \mathbf{R}, x_0) p(\mathbf{R} \mid x_0)}{\int_{SO(3)} p(y \mid \mathbf{R}', x_0) p(\mathbf{R}' \mid x_0) d\mathbf{R}'} \tag{88}$$

and noticing that $p(\mathbf{R} \mid x_0) = u_{\mathbf{R}}(\mathbf{R})$.

### B.3 THE OPTIMAL DENOISER IN THE GENERAL SETTING

The same idea and calculations from Section 3.1 hold in the general setting, where $p_x$ is arbitrary.

Using the isotropy of the Gaussian $p_\eta$:

$$l_{\text{aug}}(D) = \mathbb{E}_{\mathbf{R} \sim u_{\mathbf{R}}} \mathbb{E}_{x \sim p_x} \mathbb{E}_{\eta \sim \mathcal{N}(0, \sigma^2 \mathbb{I}_{N \times 3})} \|D(\mathbf{R}(x + \eta)) - \mathbf{R} \circ x\|^2 \tag{89}$$

$$= \mathbb{E}_{\mathbf{R} \sim u_{\mathbf{R}}} \mathbb{E}_{x \sim p_x} \mathbb{E}_{\eta \sim \mathcal{N}(0, \sigma^2 \mathbb{I}_{N \times 3})} \|D(\mathbf{R} \circ x + \eta) - \mathbf{R} \circ x\|^2 \tag{90}$$

$$= \mathbb{E}_{\mathbf{R} \sim u_{\mathbf{R}}} \mathbb{E}_{x \sim p_x} \mathbb{E}_{y \sim \mathcal{N}(\mathbf{R} \circ x, \sigma^2 \mathbb{I}_{N \times 3})} \|D(y) - \mathbf{R} \circ x\|^2 \tag{91}$$

$$= \int_{SO(3)} \int_{\mathbb{R}^{N \times 3}} \left( \int_{\mathbb{R}^{N \times 3}} \|D(y) - \mathbf{R} \circ x_0\|^2 \mathcal{N}(y; \mathbf{R} \circ x, \sigma^2 \mathbb{I}_{N \times 3}) dy \right) p_x(x) dx \, u_{\mathbf{R}}(\mathbf{R}) d\mathbf{R} \tag{92}$$

$$= \int_{\mathbb{R}^{N \times 3}} \left( \int_{\mathbb{R}^{N \times 3}} \left( \int_{SO(3)} \|D(y) - \mathbf{R} \circ x\|^2 \mathcal{N}(y; \mathbf{R} \circ x, \sigma^2 \mathbb{I}_{N \times 3}) u_{\mathbf{R}}(\mathbf{R}) d\mathbf{R} \right) p_x(x) dx \right) dy \tag{93}$$

$$= \int_{\mathbb{R}^{N \times 3}} l_{\text{aug}}(D; y, \sigma) dy \tag{94}$$

where:

$$l_{\text{aug}}(D; y, \sigma) = \int_{\mathbb{R}^{N \times 3}} \int_{SO(3)} \|D(y) - \mathbf{R} \circ x\|^2 \, \mathcal{N}(y; \mathbf{R} \circ x, \sigma^2 \mathbb{I}_{N \times 3}) \, u_{\mathbf{R}}(\mathbf{R}) d\mathbf{R} \, p_x(x) \, dx \quad (95)$$

$$= \mathbb{E}_{x \sim p_x} \mathbb{E}_{\mathbf{R} \sim u_{\mathbf{R}}} \|D(y) - \mathbf{R} \circ x\|^2 \, \mathcal{N}(y; \mathbf{R} \circ x, \sigma^2 \mathbb{I}_{N \times 3}) \quad (96)$$

Note that $l_{\text{aug}}(D; y)$ is non-negative for all $y$. Thus, the optimal denoiser $D^*$ should minimize $l_{\text{aug}}(D; y)$ for each possible $y$. Taking the gradient of $l_{\text{aug}}(D; y)$ with respect to $D(y)$ and setting it to 0:

$$\nabla_{D(y)} l_{\text{aug}}(D; y, \sigma) = 0 \quad (97)$$

$$\implies \mathbb{E}_{x \sim p_x} \mathbb{E}_{\mathbf{R} \sim u_{\mathbf{R}}} \left[ 2(D^*(y) - \mathbf{R} \circ x) \mathcal{N}(y; \mathbf{R} \circ x, \sigma^2 \mathbb{I}_{N \times 3}) \right] = 0 \quad (98)$$

$$\implies D^*(y) = \frac{\mathbb{E}_{x \sim p_x} \mathbb{E}_{\mathbf{R} \sim u_{\mathbf{R}}} [\mathbf{R} \circ x \, \mathcal{N}(y; \mathbf{R} \circ x, \sigma^2 \mathbb{I}_{N \times 3})]}{\mathbb{E}_{x \sim p_x} \mathbb{E}_{\mathbf{R} \sim u_{\mathbf{R}}} [\mathcal{N}(y; \mathbf{R} \circ x, \sigma^2 \mathbb{I}_{N \times 3})]} \quad (99)$$

which clearly specializes to Equation 81 in the single sample setting. As before, we can write this as:

$$D^*(y) = \frac{\mathbb{E}_{x \sim p_x} \mathbb{E}_{\mathbf{R} \sim u_{\mathbf{R}}} \mathbf{R} \circ x \, \mathcal{N}(y; \mathbf{R} \circ x, \sigma^2 \mathbb{I}_{N \times 3})}{\mathbb{E}_{x \sim p_x} \mathbb{E}_{\mathbf{R} \sim u_{\mathbf{R}}} \mathcal{N}(y; \mathbf{R} \circ x, \sigma^2 \mathbb{I}_{N \times 3})} \quad (100)$$

$$= \frac{\int_{SO(3)} \int_{\mathbb{R}^{N \times 3}} \mathbf{R} \circ x \, \mathcal{N}(y; \mathbf{R} \circ x, \sigma^2 \mathbb{I}_{N \times 3}) p_x(x) u_{\mathbf{R}}(\mathbf{R}) dx d\mathbf{R}}{\int_{SO(3)} \int_{\mathbb{R}^{N \times 3}} \mathcal{N}(y; \mathbf{R}' x', \sigma^2 \mathbb{I}_{N \times 3}) p_x(x') u_{\mathbf{R}}(\mathbf{R}') dx' d\mathbf{R}'} \quad (101)$$

$$= \int_{SO(3)} \int_{\mathbb{R}^{N \times 3}} \mathbf{R} \circ x \, \frac{\mathcal{N}(y; \mathbf{R} \circ x, \sigma^2 \mathbb{I}_{N \times 3}) p_x(x) u_{\mathbf{R}}(\mathbf{R}) dx d\mathbf{R}}{\int_{SO(3)} \int_{\mathbb{R}^{N \times 3}} \mathcal{N}(y; \mathbf{R}' x', \sigma^2 \mathbb{I}_{N \times 3}) p_x(x') u_{\mathbf{R}}(\mathbf{R}') dx' d\mathbf{R}'} \quad (102)$$

$$= \int_{SO(3)} \int_{\mathbb{R}^{N \times 3}} \mathbf{R} \circ x \, p(x, \mathbf{R} \mid y, \sigma) dx d\mathbf{R} \quad (103)$$

$$= \mathbb{E}_{x, \mathbf{R} \sim p(x, \mathbf{R} \mid y, \sigma)} [\mathbf{R} \circ x] \quad (104)$$

$$= \mathbb{E}_{x \sim p(x \mid y, \sigma)} [\mathbb{E}_{\mathbf{R} \sim p(\mathbf{R} \mid y, x, \sigma)} [\mathbf{R} \circ x]] \quad (105)$$

$$= \mathbb{E}_{x \sim p(x \mid y, \sigma)} [D^*(y; x)] \quad (106)$$

as the conditional probability distribution $p(x, \mathbf{R} \mid y, \sigma)$ over both point clouds $x$ and rotations $\mathbf{R}$ is:

$$p(x, \mathbf{R} \mid y, \sigma) = \frac{p(y \mid \mathbf{R}, x, \sigma) p(x) p(\mathbf{R})}{\int_x \int_{\mathbf{R}} p(y \mid \mathbf{R}, x, \sigma) p(x) p(\mathbf{R})} \quad (107)$$

$$= \frac{\mathcal{N}(y; \mathbf{R} \circ x, \sigma^2 \mathbb{I}_{N \times 3}) p_x(x) u_{\mathbf{R}}(\mathbf{R})}{\int_{SO(3)} \int_{\mathbb{R}^{N \times 3}} \mathcal{N}(y; \mathbf{R}' x', \sigma^2 \mathbb{I}_{N \times 3}) p_x(x') u_{\mathbf{R}}(\mathbf{R}') dx' d\mathbf{R}'} \quad (108)$$

Note that the marginal distribution over $\mathbf{R}$ under $p(x, \mathbf{R} \mid y, \sigma)$ is indeed $p(\mathbf{R} \mid y, x, \sigma)$ as derived in Equation 10. Note that:

$$p(\mathbf{R} \mid y, x, \sigma) = \frac{\mathcal{N}(y; \mathbf{R} \circ x, \sigma^2 \mathbb{I}_{N \times 3}) u_{\mathbf{R}}(\mathbf{R})}{\int_{SO(3)} \mathcal{N}(y; \mathbf{R}' x, \sigma^2 \mathbb{I}_{N \times 3}) u_{\mathbf{R}}(\mathbf{R}') d\mathbf{R}'} \quad (109)$$

because:

$$p(y \mid \mathbf{R}, x, \sigma) = \mathcal{N}(y; \mathbf{R} \circ x, \sigma^2 \mathbb{I}_{N \times 3}) \quad (110)$$

and Bayes' rule:

$$p(\mathbf{R} \mid y, x) = \frac{p(y, \mathbf{R} \mid x)}{p(y \mid x)} = \frac{p(y \mid \mathbf{R}, x, \sigma) p(\mathbf{R} \mid x)}{\int_{SO(3)} p(y \mid \mathbf{R}', x) p(\mathbf{R}' \mid x) d\mathbf{R}'} \quad (111)$$

and noticing that $p(\mathbf{R} \mid x) = u_{\mathbf{R}}(\mathbf{R})$.

### B.4 CONNECTION TO THE MATRIX FISHER DISTRIBUTION

Here, we show that $p(\mathbf{R} \mid y; x_0, \sigma)$ belongs to the family of Matrix Fisher distributions:

$$p(\mathbf{R} \mid y; x_0, \sigma) \propto \mathcal{N}(y; \mathbf{R} \circ x_0, \sigma^2 \mathbb{I}_{N\times 3}) u_{\mathbf{R}}(\mathbf{R}) \tag{112}$$

$$\propto \exp\left(-\frac{\|y - \mathbf{R} \circ x_0\|^2}{2\sigma^2}\right) \tag{113}$$

$$\propto \exp\left(-\frac{\|y\|^2 + \|\mathbf{R} \circ x_0\|^2 - 2\,\mathrm{Tr}[y^T(\mathbf{R} \circ x_0)]}{2\sigma^2}\right) \tag{114}$$

$$\propto \exp\left(-\frac{\|y\|^2 + \|x_0\|^2 - 2\,\mathrm{Tr}[y^T(\mathbf{R} \circ x_0)]}{2\sigma^2}\right) \tag{115}$$

$$\propto \exp\left(\frac{\mathrm{Tr}[y^T(\mathbf{R} \circ x_0)]}{\sigma^2}\right) \tag{116}$$

$$\propto \exp\left(\frac{\mathrm{Tr}[y^T(x_0 \mathbf{R}^T)]}{\sigma^2}\right) \tag{117}$$

$$\propto \exp\left(\frac{\mathrm{Tr}[\mathbf{R}^T y^T x_0]}{\sigma^2}\right) \tag{118}$$

$$\propto \exp\left(\frac{\mathrm{Tr}[(y^T x_0)^T \mathbf{R}]}{\sigma^2}\right) \tag{119}$$

$$\propto \exp\left(\mathrm{Tr}\left[\left(\frac{y^T x_0}{\sigma^2}\right)^T \mathbf{R}\right]\right) \tag{120}$$

Hence, $p(\mathbf{R} \mid y; x_0, \sigma) = \mathrm{MF}(\mathbf{R}; \frac{y^T x_0}{\sigma^2})$.

### B.5 THE OPTIMAL CONDITIONAL DENOISER IS $SO(3)$-EQUIVARIANT

Here, we show the that the optimal denoiser $D^*$ is indeed *equivariant* under rotations of $y$.

For an arbitrary rotation $\mathbf{R}'$:

$$D^*(\mathbf{R}'y; x_0, \sigma) = \frac{\mathbb{E}_{\mathbf{R} \sim u_{\mathbf{R}}}[\mathbf{R} \circ x_0\, \mathcal{N}(\mathbf{R}'y; \mathbf{R} \circ x_0, \sigma^2 \mathbb{I}_{N\times 3})]}{\mathbb{E}_{\mathbf{R} \sim u_{\mathbf{R}}}[\mathcal{N}(\mathbf{R}'y; \mathbf{R} \circ x_0, \sigma^2 \mathbb{I}_{N\times 3})]} \tag{121}$$

$$= \frac{\mathbb{E}_{\mathbf{R} \sim u_{\mathbf{R}}}[\mathbf{R} \circ x_0\, \mathcal{N}(y; (\mathbf{R}')^{-1}\mathbf{R} \circ x_0, \sigma^2 \mathbb{I}_{N\times 3})]}{\mathbb{E}_{\mathbf{R} \sim u_{\mathbf{R}}}[\mathcal{N}(y; (\mathbf{R}')^{-1}\mathbf{R} \circ x_0, \sigma^2 \mathbb{I}_{N\times 3})]} \tag{122}$$

$$= \frac{\mathbb{E}_{\mathbf{R} \sim u_{\mathbf{R}}}[\mathbf{R}'((\mathbf{R}')^{-1}\mathbf{R})x_0\, \mathcal{N}(y; ((\mathbf{R}')^{-1}\mathbf{R})x_0, \sigma^2 \mathbb{I}_{N\times 3})]}{\mathbb{E}_{\mathbf{R} \sim u_{\mathbf{R}}}[\mathcal{N}(y; ((\mathbf{R}')^{-1}\mathbf{R})x_0, \sigma^2 \mathbb{I}_{N\times 3})]} \tag{123}$$

$$= \frac{\mathbb{E}_{\mathbf{R}'' \sim u_{\mathbf{R}}}[\mathbf{R}'\mathbf{R}''x_0\, \mathcal{N}(y; \mathbf{R}''x_0, \sigma^2 \mathbb{I}_{N\times 3})]}{\mathbb{E}_{\mathbf{R}'' \sim u_{\mathbf{R}}}[\mathcal{N}(y; \mathbf{R}''x_0, \sigma^2 \mathbb{I}_{N\times 3})]} \tag{124}$$

$$= \mathbf{R}'\frac{\mathbb{E}_{\mathbf{R}'' \sim u_{\mathbf{R}}}[\mathbf{R}''x_0\, \mathcal{N}(y; \mathbf{R}''x_0, \sigma^2 \mathbb{I}_{N\times 3})]}{\mathbb{E}_{\mathbf{R}'' \sim u_{\mathbf{R}}}[\mathcal{N}(y; \mathbf{R}''x_0, \sigma^2 \mathbb{I}_{N\times 3})]} \tag{125}$$

$$= \mathbf{R}'D^*(y; x_0, \sigma) \tag{126}$$

where we used the fact that $u_{\mathbf{R}}$ is uniform so $\mathbf{R}'' \equiv (\mathbf{R}')^{-1}\mathbf{R}$ is also distributed as $u_{\mathbf{R}}$, by the invariance of the Haar measure.

Next, we show that the optimal conditional denoiser $D^*$ is indeed *invariant* under rotations of conditioning $x_0$. For an arbitrary rotation $\mathbf{R}'$:

$$D^*(y; \mathbf{R}'x_0, \sigma) = \frac{\mathbb{E}_{\mathbf{R}\sim u_{\mathbf{R}}}[\mathbf{R} \circ \mathbf{R}'x_0 \, \mathcal{N}(y; \mathbf{R} \circ \mathbf{R}'x_0, \sigma^2\mathbb{I}_{N\times3})]}{\mathbb{E}_{\mathbf{R}\sim u_{\mathbf{R}}}[\mathcal{N}(y; \mathbf{R} \circ \mathbf{R}'x_0, \sigma^2\mathbb{I}_{N\times3})]} \tag{127}$$

$$= \frac{\mathbb{E}_{\mathbf{R}\sim u_{\mathbf{R}}}[(\mathbf{R}\mathbf{R}')x_0 \, \mathcal{N}(y; (\mathbf{R}\mathbf{R}') \circ x_0, \sigma^2\mathbb{I}_{N\times3})]}{\mathbb{E}_{\mathbf{R}\sim u_{\mathbf{R}}}[\mathcal{N}(y; (\mathbf{R}\mathbf{R}') \circ x_0, \sigma^2\mathbb{I}_{N\times3})]} \tag{128}$$

$$= \frac{\mathbb{E}_{\mathbf{R}''\sim u_{\mathbf{R}}}[\mathbf{R}''x_0 \, \mathcal{N}(y; \mathbf{R}'' \circ x_0, \sigma^2\mathbb{I}_{N\times3})]}{\mathbb{E}_{\mathbf{R}\sim u_{\mathbf{R}}}[\mathcal{N}(y; \mathbf{R}'' \circ x_0, \sigma^2\mathbb{I}_{N\times3})]} \tag{129}$$

$$= D^*(y; x_0, \sigma) \tag{130}$$

where we again used the fact that $u_{\mathbf{R}}$ is uniform so $\mathbf{R}'' \equiv \mathbf{R}\mathbf{R}'$ is also distributed as $u_{\mathbf{R}}$, by the invariance of the Haar measure.

## B.6 ROTATIONAL ALIGNMENT COMMUTES WITH ROTATIONAL AUGMENTATION

Here, we show that alignment commutes with the rotation $\mathbf{R}_{\text{aug}}$ used for augmentation. In particular, the alignment procedure returns $\mathbf{R}^*(\mathbf{R}_{\text{aug}} \circ y, \mathbf{R}_{\text{aug}} \circ x) = \mathbf{R}_{\text{aug}}\mathbf{R}^*(y, x)\mathbf{R}_{\text{aug}}^T$. This is because:

$$\mathbf{R}^*(\mathbf{R}_{\text{aug}} \circ y, \mathbf{R}_{\text{aug}} \circ x) = \underset{\mathbf{R}\in SO(3)}{\operatorname{argmin}} \|\mathbf{R}_{\text{aug}} \circ y - \mathbf{R}\mathbf{R}_{\text{aug}} \circ x\| \tag{131}$$

$$= \underset{\mathbf{R}\in SO(3)}{\operatorname{argmin}} \|y - \mathbf{R}_{\text{aug}}^T \mathbf{R}\mathbf{R}_{\text{aug}} \circ x\| \tag{132}$$

$$\implies \mathbf{R}_{\text{aug}}^T\mathbf{R}^*(\mathbf{R}_{\text{aug}} \circ y, \mathbf{R}_{\text{aug}} \circ x)\mathbf{R}_{\text{aug}} = \mathbf{R}^*(y, x) \tag{133}$$

$$\implies \mathbf{R}^*(\mathbf{R}_{\text{aug}} \circ y, \mathbf{R}_{\text{aug}} \circ x) = \mathbf{R}_{\text{aug}}\mathbf{R}^*(y, x)\mathbf{R}_{\text{aug}}^T \tag{134}$$

Thus, on aligning $\mathbf{R}_{\text{aug}} \circ x$ to $\mathbf{R}_{\text{aug}} \circ y$, we get $\mathbf{R}^*(\mathbf{R}_{\text{aug}}y, \mathbf{R}_{\text{aug}}x) \circ (\mathbf{R}_{\text{aug}}x) = \mathbf{R}_{\text{aug}}\mathbf{R}^*(y, x)\mathbf{R}_{\text{aug}}^T\mathbf{R}_{\text{aug}} \circ x = \mathbf{R}_{\text{aug}}\mathbf{R}^*(y, x) \circ x$.

## B.7 AVERAGING AN ESTIMATOR INDUCES AN EQUIVALENT MATCHING LOSS

Here, we show that averaging an estimator $D_{\text{est}}$ gives us an equivalent matching loss from the perspective of minimization with respect to $D$. Formally, for any estimator $D_{\text{est}}$ we have:

$$l_{\text{est}}(D; \mathbb{E}_{\mathbf{R}\sim p(\mathbf{R} \mid y,x,\sigma)}[D_{\text{est}}]) = l_{\text{est}}(D; D_{\text{est}}) + C \tag{135}$$

where $C$ is a constant that does not depend on $D$. We have:

$l_{\text{est}}(D; D_{\text{est}})$

$$= \mathbb{E}_{\sigma\sim p_\sigma}\mathbb{E}_{y\sim p(y|\sigma)}\mathbb{E}_{x\sim p(x \mid y,\sigma)}\mathbb{E}_{\mathbf{R}\sim p(\mathbf{R} \mid y,x,\sigma)}[\|D(y;\sigma) - D_{\text{est}}(y; x, \mathbf{R}, \sigma)\|^2] \tag{136}$$

$$= \mathbb{E}_\sigma\mathbb{E}_y\mathbb{E}_x\mathbb{E}_{\mathbf{R}}[\|D(y;\sigma) - D_{\text{est}}(y; x, \mathbf{R}, \sigma)\|^2] \tag{137}$$

$$= \mathbb{E}_\sigma\mathbb{E}_y\mathbb{E}_x\mathbb{E}_{\mathbf{R}}[\|D(y;\sigma) - \mathbb{E}_{\mathbf{R}}[D_{\text{est}}(y; x, \mathbf{R}, \sigma)] + \mathbb{E}_{\mathbf{R}}[D_{\text{est}}(y; x, \mathbf{R}, \sigma)] - D_{\text{est}}(y; x, \mathbf{R}, \sigma)\|^2] \tag{138}$$

$$= \mathbb{E}_\sigma\mathbb{E}_y\mathbb{E}_x\mathbb{E}_{\mathbf{R}}[\|D(y;\sigma) - \mathbb{E}_{\mathbf{R}}[D_{\text{est}}(y; x, \mathbf{R}, \sigma)]\|^2 + \|\mathbb{E}_{\mathbf{R}}[D_{\text{est}}(y; x, \mathbf{R}, \sigma)] - D_{\text{est}}(y; x, \mathbf{R}, \sigma)\|^2$$
$$+ 2(D(y;\sigma) - \mathbb{E}_{\mathbf{R}}[D_{\text{est}}(y; x, \mathbf{R}, \sigma)])^\top(\mathbb{E}_{\mathbf{R}}[D_{\text{est}}(y; x, \mathbf{R}, \sigma)] - D_{\text{est}}(y; x, \mathbf{R}, \sigma))] \tag{139}$$

where we omit the explicit distributions for clarity. Now, focusing on the last term:

$$\mathbb{E}_{\mathbf{R}}[2(D(y;\sigma) - \mathbb{E}_{\mathbf{R}}[D_{\text{est}}(y; x, \mathbf{R}, \sigma)])^\top(\mathbb{E}_{\mathbf{R}}[D_{\text{est}}(y; x, \mathbf{R}, \sigma)] - D_{\text{est}}(y; x, \mathbf{R}, \sigma))] \tag{140}$$

$$= 2(D(y;\sigma) - \mathbb{E}_{\mathbf{R}}[D_{\text{est}}(y; x, \mathbf{R}, \sigma)])^\top\mathbb{E}_{\mathbf{R}}[(\mathbb{E}_{\mathbf{R}}[D_{\text{est}}(y; x, \mathbf{R}, \sigma)] - D_{\text{est}}(y; x, \mathbf{R}, \sigma))] \tag{141}$$

$$= 2(D(y;\sigma) - \mathbb{E}_{\mathbf{R}}[D_{\text{est}}(y; x, \mathbf{R}, \sigma)])^\top(\mathbb{E}_{\mathbf{R}}[D_{\text{est}}(y; x, \mathbf{R}, \sigma)] - \mathbb{E}_{\mathbf{R}}[D_{\text{est}}(y; x, \mathbf{R}, \sigma)]) \tag{142}$$

$$= 2(D(y;\sigma) - \mathbb{E}_{\mathbf{R}}[D_{\text{est}}(y; x, \mathbf{R}, \sigma)])^\top\mathbf{0} \tag{143}$$

$$= \mathbf{0}. \tag{144}$$

as the first term in the product is a constant with respect to $\mathbf{R}$. Thus,

$$l_{\text{est}}(D; D_{\text{est}}) \tag{145}$$

$$= \mathbb{E}_\sigma \mathbb{E}_y \mathbb{E}_x \mathbb{E}_{\mathbf{R}}[\|D(y; \sigma) - \mathbb{E}_{\mathbf{R}}[D_{\text{est}}(y; x, \mathbf{R}, \sigma)]\|^2 + \|\mathbb{E}_{\mathbf{R}}[D_{\text{est}}(y; x, \mathbf{R}, \sigma)] - D_{\text{est}}(y; x, \mathbf{R}, \sigma)\|^2] \tag{146}$$

$$= l_{\text{est}}(D; \mathbb{E}_{\mathbf{R}}[D_{\text{est}}]) + \underbrace{\mathbb{E}_\sigma \mathbb{E}_y \mathbb{E}_x \mathbb{E}_{\mathbf{R}}[\|\mathbb{E}_{\mathbf{R}}[D_{\text{est}}(y; x, \mathbf{R}, \sigma)] - D_{\text{est}}(y; x, \mathbf{R}, \sigma)\|^2]}_{\text{independent of } D}. \tag{147}$$

as claimed.

## C FOR A GENERAL MATRIX FISHER DISTRIBUTION

We note that the partition function $Z(F) = \int_{SO(3)} \exp(\text{Tr}[F^\top \mathbf{R}]) d\mathbf{R}$ gives all the information we need. In particular, taking derivatives of $Z(F)$ allows us to calculate any necessary moments. For example,

$$\mathbb{E}_{\mathbf{R} \sim \text{MF}(\mathbf{R}; F)}[\mathbf{R}] = \frac{\int_{SO(3)} \mathbf{R} \exp(\text{Tr}[F^\top \mathbf{R}]) d\mathbf{R}}{Z(F)} = \frac{\frac{d}{dF} Z(F)}{Z(F)} = \frac{d}{dF} \ln Z(F) \tag{148}$$

using the trace derivative identity: $\frac{d}{dF} \text{Tr}[F^\top \mathbf{R}] = \mathbf{R}$.

Now, we remove the explicit $\sigma$-dependence of $F = \frac{y^\top x}{\sigma^2}$, by defining $F' = y^\top x$ and $\lambda = \frac{1}{\sigma^2}$, so that $F = \lambda F'$. We are interested in computing $\mathbb{E}_{\mathbf{R} \sim \text{MF}(\mathbf{R}; \lambda F')}[\mathbf{R}]$ in the limit of $\lambda \to \infty$.

Recall that we can always factorize $F' = USV^\top$ where $U, V \in SO(3)$ and $S = \text{diag}[s_1, s_2, s_3]$ where $s_1 \geq s_2 \geq |s_3|$, by the Singular Value Decomposition. In particular, we see that:

$$\text{Tr}[F'^\top \mathbf{R}] = \text{Tr}[VS^\top U^\top \mathbf{R}] = \text{Tr}[S^\top U^\top \mathbf{R} V] = \text{Tr}[S^\top (U^\top \mathbf{R} V)] \tag{149}$$

by the cyclic property of the trace. Since $U, V \in SO(3)$ it follows that $Z(F') = Z(S)$ (by change of variables $\mathbf{R} \to U^\top \mathbf{R} V$) so we can restrict our calculations to the diagonal case .

Laplace's method provides a powerful tool to expand integrals of sharply peaked functions. The key idea is that only the neighborhood of a sharp peak has significant contributions and that such a region can be approximated with a Gaussian distribution. In our case, we seek to apply this method to $Z(\lambda S)$ in the limit of $\lambda \to \infty$.

As $S$ is diagonal, it is easy to see that $\text{argmax}_{\mathbf{R} \sim SO(3)}[\lambda \text{Tr}[S\mathbf{R}]] = \mathbb{I}_{3 \times 3}$. Hence, a natural parameterization to use is the exponential map expansion of $SO(3)$ which expands around the identity. In this chart, we have parameters $\boldsymbol{\theta} = (\theta_x, \theta_y, \theta_z)$ and our rotation is given by

$$\mathbf{R}(\theta_x, \theta_y, \theta_z) = \exp(\theta_x R_x + \theta_y R_y + \theta_z R_z) \tag{150}$$

where $R_x, R_y, R_z$ are the generators of $x, y, z$ rotations. In this parameterization, the Haar measure can be found to be

$$\mu(\boldsymbol{\theta}) d\theta_x d\theta_y d\theta_z = \frac{1 - \cos(\|\boldsymbol{\theta}\|)}{4\pi^2 \|\boldsymbol{\theta}\|^2} d\theta_x d\theta_y d\theta_z. \tag{151}$$

Next, we would like to expand the argument $\lambda A(\boldsymbol{\theta}, S) = \lambda \text{Tr}[S\mathbf{R}(\boldsymbol{\theta})]$ around $\boldsymbol{\theta} = 0$. Because this is maximized at $\mathbf{R} = \mathbb{I}$ so $\boldsymbol{\theta} = 0$, we obtain an expression of the form:

$$A(\boldsymbol{\theta}, S) = A_0(S) + \sum_{ij} A_{2,ij}(S) \theta_i \theta_j + \sum_{ijk} A_{3,ijk}(S) \theta_i \theta_j \theta_k + \dots \tag{152}$$

where the indices $i, j, k$ run over $\{x, y, z\}$.

Define $B(\boldsymbol{\theta}, S, \lambda) \equiv \exp(\lambda(A(\boldsymbol{\theta}, S) - A_0(S) - \sum_{ij} A_{2,ij}(S) \theta_i \theta_j))$. We can then write:

$$\exp(\lambda A(\boldsymbol{\theta}, S)) = \exp(\lambda A_0(S) + \sum_{ij} \lambda A_{2,ij}(S) \theta_i \theta_j) \exp(\lambda(A(\boldsymbol{\theta}, S) - A_0(S) - \sum_{ij} A_{2,ij}(S) \theta_i \theta_j))$$

$$= \exp(\lambda A_0(S) + \sum_{ij} \lambda A_{2,ij}(S) \theta_i \theta_j) B(\boldsymbol{\theta}, S, \lambda).$$

To integrate over all of $SO(3)$, we simply need to integrate over the domain $\{|\boldsymbol{\theta}| < \pi\}$. Hence, we would like to evaluate:

$$\int_{|\boldsymbol{\theta}|<\pi} e^{\lambda A_0(S) + \sum_{ij} \lambda A_{2,ij}(S)\theta_i\theta_j} B(\boldsymbol{\theta}, S, \lambda)\mu(\boldsymbol{\theta})\mathrm{d}\boldsymbol{\theta}. \tag{153}$$

Since $\boldsymbol{\theta} = 0$ is a local maxima, $A_{2,ij}(S)$ must be negative definite so the exponential component can be interpreted as a Gaussian. As $\lambda \to \infty$, this Gaussian becomes increasingly peaked; therefore, only the neighborhood around $\boldsymbol{\theta} = 0$ matters. Hence, we can expand $B(\boldsymbol{\theta}, S, \lambda)\mu(\boldsymbol{\theta})$ around $0$ to get:

$$B(\boldsymbol{\theta}, S, \lambda)\mu(\boldsymbol{\theta}) = M_0 + \sum_{ij} M_{2,ij}\theta_i\theta_j + \sum_{ijk} M_{3,ijk}(S, \lambda)\theta_i\theta_j\theta_k + \dots \tag{154}$$

where the contributions up to second order can only come from the expansion of the measure (which has no $S$ or $\lambda$ dependence), and there is no first order term since the measure is symmetric around $0$. Hence, Equation 153 becomes:

$$\int_{|\boldsymbol{\theta}|<\pi} e^{\lambda A_0(S) + \sum_{ij} \lambda A_{2,ij}(S)\theta_i\theta_j} \left( M_0 + \sum_{ij} M_{2,ij}\theta_i\theta_j + \sum_{ijk} M_{3,ijk}(S, \lambda)\theta_i\theta_j\theta_k + \dots \right) \mathrm{d}\boldsymbol{\theta}. \tag{155}$$

Finally, we note that as $\lambda \to \infty$, the Gaussian part has an increasingly sharp peak. Thus, for the expansion terms in Equation 155, the boundaries of integration matter increasingly less. Hence, replacing the domain $\{|\boldsymbol{\theta}| < \pi\}$ with the larger domain $\{\boldsymbol{\theta} \in \mathbb{R}^3\}$ gives us a good approximation for each of the expanded terms, but also gives us Gaussian integrals which are analytically evaluable.

Finally, we obtain an expression of the form:

$$Z(\lambda S) = N(S, \lambda) \left( 1 + L_1(S)\frac{1}{\lambda} + L_2(S)\frac{1}{\lambda^2} + L_3(S)\frac{1}{\lambda^3} + \dots \right) \tag{156}$$

where $N(S, \lambda)$ is a normalization term. The corresponding expected rotation can be computed as:

$$\mathbb{E}_{\mathbf{R}\sim\mathrm{MF}(\mathbf{R};\lambda S)}[\mathbf{R}] = \frac{1}{\lambda}\,\mathrm{diag}\left[ \frac{\partial \ln Z(\lambda S)}{\partial s_1}, \frac{\partial \ln Z(\lambda S)}{\partial s_1}, \frac{\partial \ln Z(\lambda S)}{\partial s_3} \right]$$

$$= \mathbb{I} + C_1(S)\frac{1}{\lambda} + C_2(S)\frac{1}{\lambda^2} + \dots \tag{157}$$

For an arbitrary $F' = USV^\top$, we would then have:

$$\mathbb{E}_{\mathbf{R}\sim\mathrm{MF}(\mathbf{R};\lambda F')}[\mathbf{R}] = U\mathbb{E}_{\mathbf{R}\sim\mathrm{MF}(\mathbf{R};\lambda S)}[\mathbf{R}]V^\top. \tag{158}$$

## D  ADDITIONAL RESULTS IN PRACTICE

Here, we report the RMSD after alignment for the same training runs in Figure 4 and Figure 5. Note that our estimators are not optimal for this metric; indeed, they minimize the deviation to the optimal denoiser, not to the aligned ground truth $x$. The complication is that computing the optimal denoiser is not practical in a training setup.

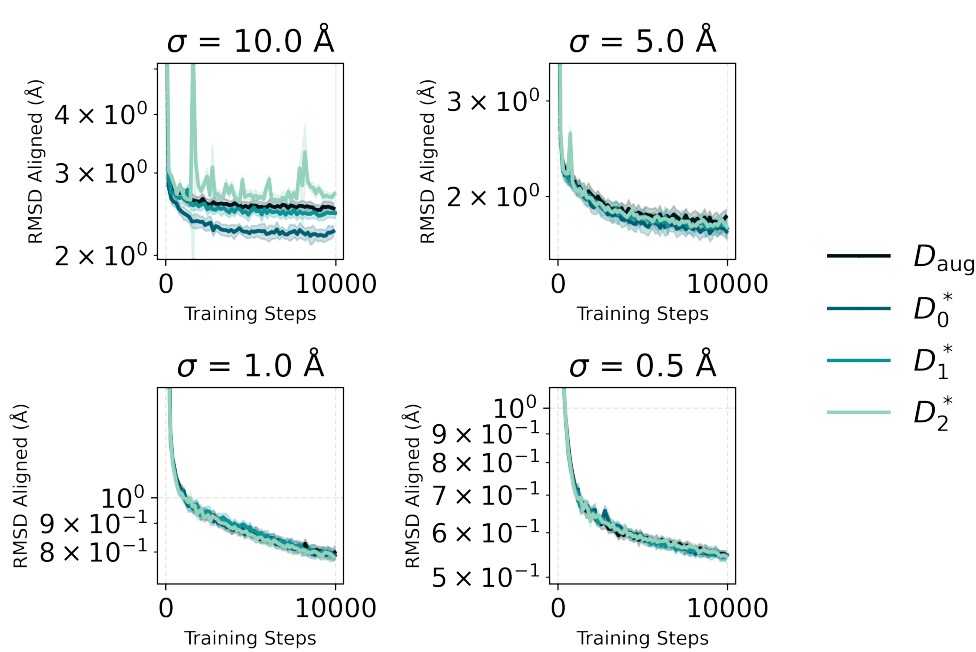

Figure 6: Training progress for the MLP, as measured by RMSD to ground-truth $x$ after alignment, when trained using $D_{\text{aug}}$, $D_0^*$, $D_1^*$ and $D_2^*$. $x$ is sampled from all $50000$ frames of a molecular dynamics simulation for the AEQN peptide.

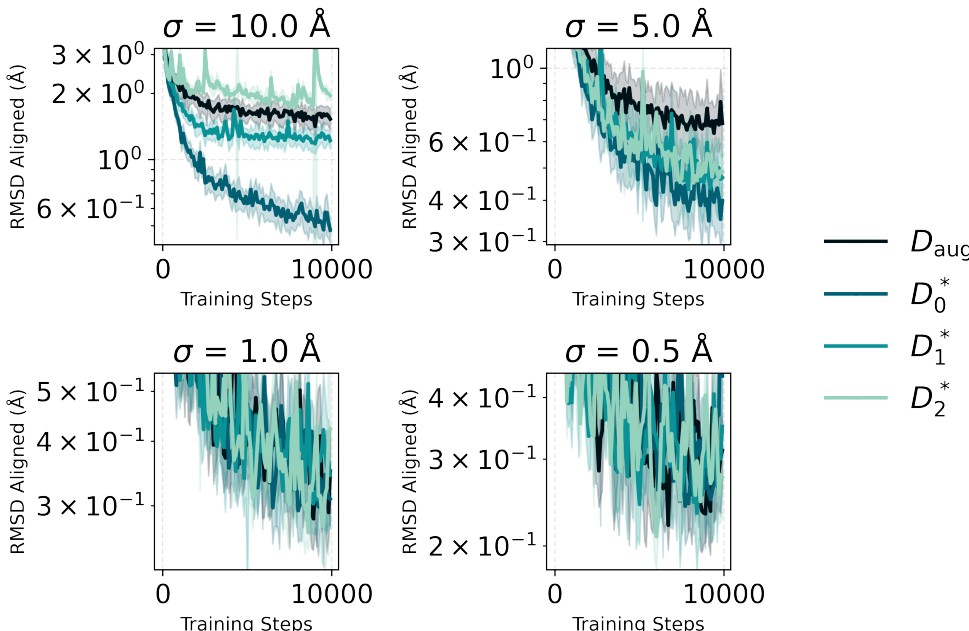

Figure 7: Training progress for the MLP, as measured by RMSD to ground-truth $x$ after alignment, when trained using $D_{\text{aug}}$, $D_0^*$, $D_1^*$ and $D_2^*$. $x$ is fixed as the first frame in the molecular dynamics simulation for the AEQN peptide.

