# OpenReview forum: "Matching the Optimal Denoiser in Point Cloud Diffusion with (Improved) Rotational Alignment"
_ICLR.cc/2026/Conference — Submitted to ICLR 2026_

### Official Review · Reviewer_wydx · 2025-10-28

**Soundness:** 2
**Presentation:** 1
**Contribution:** 1
**Rating:** 2
**Confidence:** 3

**Summary:**

This paper studies enforcing rotational symmetry in point cloud diffusion, in particular aiming to explain an alignment step performed during training the denoiser aimed to enforce rotational symmetry. It then shows that alignment arises from approximating the distribution of the optimal rotationally equivariant denoiser, and proposes methods of improving alignment by computing additional terms.

Although the connection found in this paper between rotational alignment and the optimal denoiser is interesting, in my opinion it is too small a contribution, especially giving the inconclusive experimental results.

**Strengths:**

This paper demonstrates a principled way of deriving the alignment step used to train rotationally equivariant diffusion models for point-cloud data.

**Weaknesses:**

1. This paper is written in a stream-of-consciousness manner that makes it hard for a reader to figure out its motivation, results and conclusions.
2. A main motivation of this paper is to reduce the bias introduced in an alignment step for point-cloud registration before computing the denoising loss, but it is unclear whether alignment is indeed a crucial step and what (if any) biases it does introduce.
3. The experimental results for training diffusion models using additional terms of the approximation seem to be inconclusive; additional terms do not significantly improve over the baseline.

**Questions:**

1. What is the definition of a perfect denoiser? It doesn't seem to be explicitly stated. The definition that the proof in the appendix is working with seems to be quite strong, as even when the data distribution consists of two deltas, perfect denoising in this sense does not seem to be possible.
2. The Kabsch alignment algorithm is also not defined in the paper, it would be helpful to state what it does and the optimization problem it is trying to solve.

---

### Official Review · Reviewer_aBUd · 2025-10-29

**Soundness:** 3
**Presentation:** 4
**Contribution:** 2
**Rating:** 2
**Confidence:** 3

**Summary:**

The paper addresses learning denoisers for diffusion models generating 3D point clouds, e.g., molecules. Previous work has added an alignment operation to the denoising target, where the denoising output is aligned to the ground truth point cloud / molecule before calculating the loss. The paper does analysis on what is the effect of this alignment, and explores alternative methods. In particular, the paper shows that the standard denoising loss for a single data point, when augmented with rotations of the original data, can be written as an expectation of the rotated 3D coordinates where the rotation matrices come from a particular matrix Fisher distribution. For the full data set, the optimal denoiser is an expectation over these expectations. This is similar to the general result saying that the optimal denoiser in diffusion models converges to $E[x_0 | x_t]$, where $x_0$ is the clean data and $x_t$ is  a particular noisy input to the denoiser at noise level $t$. The authors then show that with suitable conditions, the mode of this optimal data-conditional denoiser corresponds to the alignment method used in previous work, justifying its use as a first approximation to the optimal unbiased denoiser. The paper then goes on to derive more advanced approximations to the expectation over the matrix Fisher distribution, with one motivation being if we train directly with the expected rotation, the variance would be reduced during training compared to the regular unaligned denoising loss while being less biased. The approximation derived does not require significant additional computation compared to the method with alignment. The advanced approximations show lower numerical error compared to the true score, and preliminary results show that with real molecular data, the derived method may help compared to the simple methods.

**Strengths:**

The paper presents a number of novel contributions, and it could help with better understanding of the approximations made in the alignment method, and it could provide a basis for further theoretical work in the domain.
- The analysis of the optimal data-conditional denoiser being an expectation over the matrix-Fisher distribution is novel, as far as I am aware, and may help with reasoning about rotation-augmented point-cloud diffusion models.
- The connection between the matrix-Fisher distribution and the alignment method may be a good theoretical basis for reasoning about the effectiveness of the alignment method, and what kinds of distributional biases it may cause.
- The advanced approximations to the optimal denoiser are novel and the mathematics is quite sophisticated, possibly helping kick-start further research on even more effective approximations.
- The experiment studying the error relative to the optimal denoiser with the baselines and the advanced approximations is a good first step towards experimental validation, and shows that in some contexts, there may be benefits to the method.

**Weaknesses:**

- The main weakness of the paper is the lack of robust experimental validation. The formal numerical error experiment does validate that the mathematics works, but the tetrapeptide experiment only has initial loss curves. Unfortunately, it is not obvious enough from the mathematics what will the practical benefits of the methods described in the paper be, and as such I believe that the paper does need more experiments clearly showing the effects of the newly derived denoising objectives, and/or whatever additional insights might be connected from the mathematics to practical experimental results.

**Questions:**

Here are some attempts at brainstorming hypotheses for the practical differences between the methods described in the paper (but by all means if you do not agree and think something else is more interesting, I am happy to see that too):
1) Perhaps the alignment (as opposed to the unbiased denoiser) is useful for decreasing the curvature of the ODE? Consider the case with a single molecule: A standard diffusion model trained without data augmentation is able to generate that molecule in a single step, starting from pure noise, but the standard rotation-augmented model can not, since the optimal denoiser at the first step points towards an average of the rotations of the molecule. As such, we need inherently more steps to break the symmetries in the generated sample. With alignment it seems that the denoiser output can, in principle, converge to any rotation of the data point since it is artificially aligned with the data in the loss. The "aligned denoiser" seems to then have the same property that it can do single-step generation for a single data point data set, analogously to the optimal ODE curvature noise schedule described in [1]. Similar analysis has been done recently in the context of permutation equivariance in diffusion models [2,3].
2) I suppose this raises the question of "what exactly is the downside of alignment"? It seems to me that one issue is that the denoiser output could converge to any arbitrary rotation of the target point cloud / molecule given any input $x_t$, and they would be equivalent w.r.t. the loss. If the output converges on different types of rotations along different noise levels in the generative path, that could jerk the sample in totally different and inconsistent directions, and would not result in a good-quality sample. Perhaps it would be possible to show something like this in a toy scenario, e.g., with a single data point? Especially if the data point is highly self-symmetric, or we somehow induce the denoiser into this failure mode?
3) Another thing that comes to mind is that would it be possible to somehow characterise mathematically the distribution that the diffusion model with the aligned loss converges to, with some assumptions? E.g., I assume that rotation invariance of the final output distribution may not hold?
4) And I suppose that the new method should have the benefit of lower variance compared to the standard denoising loss without alignment. Perhaps it is possible to show the benefit of this at least with smaller batch sizes?

For now, I am starting out with a reject, but the reason is not that I think that there is anything wrong or uninteresting about the paper. I think the paper shows good promise and is interesting, but also I think that more experimental validation of the method is needed.

References:

[1] Karras et a., "Elucidating the Design Space of Diffusion-Based Generative Models", NeurIPS 2022

[2] Lawrence et al., "Improving Equivariant Networks With Probabilistic Symmetry Breaking", ICLR 2025

[3] Laabid et al., "Equivariant Denoisers Cannot Copy Graphs: Align Your Graph Diffusion Models", ICLR 2025

---

### Official Review · Reviewer_M89b · 2025-10-30

**Soundness:** 3
**Presentation:** 2
**Contribution:** 2
**Rating:** 2
**Confidence:** 4

**Summary:**

The paper presents an analytical study of how introducing rotational augmentation and an alignment stage (using the Kabsch Umeyama algorithm) affects the diffusion loss function for point cloud data, in case the $3D$ data which is insensitive to spatial rotations. They first derive the optimal denoiser for the rotationally augmented data distribution and show that the diffusion loss function with augmentation is equivalent to matching the single sample optimal denoiser using an $L_2$ loss.
The authors then demonstrate that the standard Kabsch Umeyama alignment step corresponds to the zeroth order approximation (with respect to the noise level) of the single sample optimal denoiser, providing a theoretical justification for the alignment stage.
Finally, through empirical evaluation, they show that including higher-order approximations of the optimal denoiser yields no significant improvement in the training PMSE loss across multiple noise levels, suggesting that the standard alignment is sufficient for practical training.

**Strengths:**

1) The suggested method gains its motivation from modeling molecular and protein structures, which is a worthy.
2) In general, incorporating prior knowledge into the design of machine-learning algorithms (denoisers in this case) has the potential to boost performance.
3) Sections 1-6 are fairly written and easy to follow.

**Weaknesses:**

My overall impression is that the paper is not ready for publication and should be rejected. More details are given below.
1. The main ideas presented in the paper (Sections 1-6), albeit elegant, are quite straight-forward, since everything is linear/Gaussian. Hence I can not recognize any solid theoretical contribution in the paper. Section 7 requires some technical effort (Laplace integration).
2. In general, what is the motivation to focus on weak noise perturbations, given that diffusion models operate over a full spectrum of noise levels?
3. The idea of expanding the target denoiser is also a bit incremental.
4. The above points can be justified if the improvements in training are groundbreaking, but: (a)The numerical evidence is very poor, consisting on only one experiment. (b) No evidence is presented for improvement in sample quality of models learned under their suggested scheme (i.e., generalization), which is the main motivation in training such models.
5. The writing in Section 7 lacks many definitions and explanation of calculation steps (see ”Questions”).

**Questions:**

1. Since the method is built upon the Kabsch Algorithm, it should be explained in details in the main text or at least in the Appendix for the unfamiliar reader.

2. In line.321 $R_x,R_y,R_z$ (Lie algebra generators?) are undefined.

3. In line 326 what is $B(\theta,S,\lambda)$?

4. In line 316 "It turns out that $Z(\lambda F) = Z(\lambda S)$" please explain this claim.

5. What are the differences between Figs 3, 4 and 5?

6. Beyond improved training loss, is there any improvement in trained models quality?

7. Typo in line 322 (missing parentheses).

---

### Official Review · Reviewer_hvL6 · 2025-10-30

**Soundness:** 1
**Presentation:** 2
**Contribution:** 2
**Rating:** 4
**Confidence:** 4

**Summary:**

The paper studies the optimal denoiser in rotational invariant settings that are of practical use in understanding protein structure. This is a very interesting field of study on its own. Authors present an analytical formula for the optimal denoiser in terms of the Matrix Fisher distribution.

**Strengths:**

Based on the Matrix Fisher representation, the paper develops a reasonable perturbation theory in noise scale explaining the role of Kabsch/Umeyama alignment in optimal denoiser. In addition they evaluate corrections order by order and by small scale experiments show that their approximation scheme is under control at small noise limit. This establishes the validity of the perturbation theory in a robust manner.

**Weaknesses:**

The paper does not compare the optimal denoiser against fully trained, large-scale diffusion models. In non-rotational settings, recent work (see for instance https://arxiv.org/pdf/2509.09672) shows that the optimal denoiser aligns with UNet behavior only in the large-noise regime, while at low noise it diverges. Moreover, a local linear denoiser (Wiener filter) consistently approximates trained diffusion models better than optimal denoiser. An analogous, rotationally invariant result would be valuable here, and this might be in tension with the paper’s small-noise perturbation theory. Given the limited experiments, it remains unclear what the practical takeaway is without a head to head comparisons with trained diffusion models across noise levels.

**Questions:**

Please compare the optimal denoiser and various approximations to it against a large scale, trained diffusion model across noise levels.

---

### Meta-Review · Area_Chair_2PcN · 2026-01-07

**Summary:**

The paper presents an analytical study of rotational alignment in point-cloud diffusion models, demonstrating that the commonly used Kabsch/Umeyama alignment can be interpreted as a low-noise approximation to the optimal rotationally invariant denoiser derived from a matrix Fisher distribution. While the mathematical analysis is interesting and the connection between alignment and optimal denoising is intriguing, reviewers consistently found the overall contribution to be too limited. In particular, the work remains largely theoretical and incremental, with very weak and inconclusive empirical validation. The authors did not provide a rebuttal, so none of the reviewers’ substantive concerns were addressed.

**Reviewer Concerns:**

Reviewers agreed that the paper contains some elegant derivations and could be useful for better conceptual understanding of rotational alignment in diffusion models. However, several recurring issues dominated the evaluations. The theoretical development largely operates in linear/Gaussian or small-noise regimes, making the contributions feel narrow and incremental. The motivation for focusing on weak-noise perturbations was questioned, given that diffusion models operate across a wide range of noise levels. Multiple reviewers also pointed out that key concepts (e.g., “perfect denoiser,” Kabsch alignment details, Lie algebra terms) are insufficiently defined, and that parts of the exposition are difficult to follow. There is considerable room for improvement in the paper's clarity.

**Reviewer Scores:**

The main reason for rejection is the lack of convincing experimental evidence. Experiments are extremely limited, focusing on a single small molecular system with training-loss curves only, and show little to no practical benefit from the proposed higher-order corrections over standard alignment. There are no comparisons against large-scale, trained diffusion models, no evaluation of sample quality or generalization, and no clear demonstration that the theory leads to meaningful improvements in practice. As a result, the practical takeaway is unclear: the paper itself suggests that standard alignment is already “good enough.” Without a rebuttal or additional results to address these points, the submission does not meet the bar for acceptance.

---

### Decision · Program_Chairs · 2026-01-26

Reject